# PLMTrajRec: A Scalable and Generalizable Trajectory Recovery Method with Pre-trained Language Models

**Tonglong Wei**[1]*, **Yan Lin**[2]*, **Youfang Lin**[1,3], **Shengnan Guo**[1,4]†, **Jilin Hu**[5]
**Haitao Yuan**[6], **Gao Cong**[6], **Huaiyu Wan**[1,3]

[1] School of Computer Science and Technology, Beijing Jiaotong University, China
[2] Department of Computer Science, Aalborg University, Denmark
[3] Beijing Key Laboratory of Traffic Data Mining and Embodied Intelligence, China
[4] Key Laboratory of Big Data & Artificial Intelligence in Transportation, Ministry of Education, China
[5] School of Data Science and Engineering, East China Normal University, China
[6] College of Computing and Data Science, Nanyang Technological University
{weitonglong, guoshn, yflin, hywan}@bjtu.edu.cn, lyan@cs.aau.dk,
jlhu@dase.ecnu.edu.cn, {haitao.yuan, gaocong}@ntu.edu.sg

## Abstract

Spatiotemporal trajectory data is crucial for various traffic-related applications. However, issues such as device malfunctions and network instability often result in sparse trajectories that lose detailed movement information compared to their dense counterparts. Recovering missing points in sparse trajectories is thus essential. Despite recent progress, three challenges remain. First, the lack of large-scale dense trajectory datasets hinders the training of a trajectory recovery model. Second, the varying spatiotemporal correlations in sparse trajectories make it hard to generalize across different sampling intervals. Third, extracting road conditions for missing points is non-trivial.

To address these challenges, we propose *PLMTrajRec*, a novel trajectory recovery model. It leverages the scalability of a pre-trained language model (PLM) and can effectively recover trajectories by fine-tuning with small-scale dense trajectory datasets. To handle different sampling intervals in sparse trajectories, we first convert sampling intervals and movement features into prompts for the PLM to understand. We then introduce a trajectory encoder to unify trajectories of varying intervals into a single interval. To extract road conditions for missing points, we propose an area flow-guided implicit trajectory prompt that represents traffic conditions in each region, and a road condition passing mechanism that infers the road conditions of missing points using the observed ones. Experiments on four public trajectory datasets with three sampling intervals demonstrate the effectiveness, scalability, and generalization ability of PLMTrajRec. Code is available at `https://github.com/wtl52656/PLMTrajRec`.

## 1 Introduction

Spatiotemporal trajectories are sequences of (location, timestamp) pairs that record the movement of individuals and vehicles. They play a pivotal role in various applications, such as urban planning [36, 43, 1, 19, 37], traffic management [25, 34, 47], and personalized location services [46, 33, 6, 5].

---

*Both authors contributed equally to this research.
†Corresponding author.

39th Conference on Neural Information Processing Systems (NeurIPS 2025).

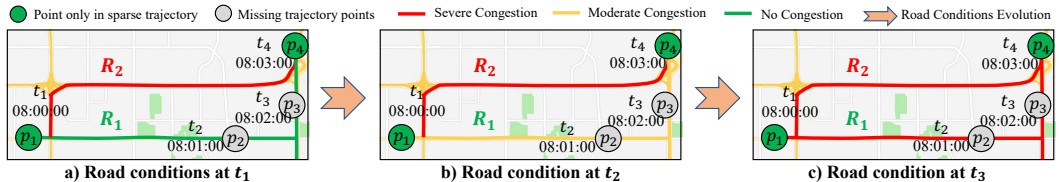

Figure 1: Impact of road conditions on route selection and movement patterns.

However, factors such as network instability, device malfunctions, or cost-saving settings often lead to sparse trajectories with large sampling intervals [26, 45]. Such sparse trajectories fail to accurately reflect movement behavior and route choices, limiting their usefulness. To address this issue, recovering the missing points in sparse trajectories is crucial for preserving trajectory completeness—a task usually referred to as *trajectory recovery*.

Two branches of methods have been proposed to tackle trajectory recovery: *free space trajectory recovery* [9, 7] and *map-matched trajectory recovery* [38, 8]. Free space methods directly predict the coordinates of missing points but do not ensure alignment with the road network, typically requiring a subsequent map-matching step. Map-matched methods, by contrast, aim to recover missing points on the road network, making them more accurately resemble real-world trajectories.

This work follows the *map-matched trajectory recovery* branch. Despite the progress, it still faces three challenges: **First, limited dense trajectory datasets.** Trajectory recovery models are data-driven and typically require large-scale pairs of sparse and dense trajectories. However, because of cost-saving settings and device malfunctions, most collected trajectories remain sparse. The limited availability of dense trajectory data makes existing models prone to overfitting and hampers their performance. **Second, difficulty in generalizing across varying sampling intervals.** Real-world sparse trajectory datasets often contain trajectories with a mixture of different sampling intervals [34]. Trajectories with different sampling intervals encompass different spatiotemporal correlations. Yet, existing works [38, 8, 26] treat different sampling intervals the same way, requiring retraining when facing trajectories with sampling intervals unseen during training, which introduces additional computational burden. **Third, non-trivial extraction of dynamic road conditions for missing points.** Road conditions of both observed and missing trajectory points are essential in facilitating more accurate trajectory recovery. For example, in Figure 1(b) and (c), knowing the conditions at $p_2$ and $p_3$ reveals that the user is gradually decelerating on $R_1$ due to congestion. If we only consider conditions at $p_1$ and $p_4$, we might infer that the user takes $R_1$ instead of $R_2$ (Figure 1(a)), but not the finer details of the movement. Yet, since the missing points in a sparse trajectory are unknown, it is non-trivial to extract the road conditions at their exact locations.

To tackle these challenges, we propose *Pre-trained Language Model for Trajectory Recovery* (**PLMTrajRec**). Drawing inspiration from [21, 15], which demonstrate that PLM possess broader general knowledge and mitigate the lack of domain-specific datasets, we integrate a PLM to enable high-performance trajectory recovery by fine-tuning on small-scale dense trajectory datasets (Challenge 1). To handle sparse trajectories with varying sampling intervals (Challenge 2), we introduce an interval and feature-guided (IF-guided) *explicit trajectory prompt*. It incorporates both sampling intervals and movement features into a prompt for the PLM, helping the model extract information from these features. We also introduce an *interval-aware trajectory embedder* to standardize different sampling intervals and learn their spatiotemporal correlations. To infer road conditions for missing points (Challenge 3), we design an area flow-guided (AF-guided) *implicit trajectory prompt* that gathers traffic flows in each region. We also present a *road condition passing mechanism* that uses road conditions from nearby observed points to estimate those of the missing points. We conduct extensive experiments on four real-world datasets, each with three sampling intervals, showing that PLMTrajRec achieves superior performance in effectiveness, scalability, and generalizability.

## 2 Related Work

**Trajectory recovery** aims to reconstruct missing points from sparse trajectories. Based on whether the road network is considered, it can be divided into two types: free-space recovery and map-matched recovery. Free-space recovery directly restores missing GPS coordinates. Traditional methods often

use predefined rules [4, 28, 12, 9] or assume Markov transitions between points [2, 48]. They are limited in capturing the global spatial-temporal dependencies essential for accurate trajectory recovery. Recent deep learning approaches improve recovery performance by modeling complex spatiotemporal patterns [40, 41], including sequence-based models [32, 39, 7, 27] and graph-based methods [30]. However, these methods typically require a separate map-matching step to align the recovered trajectory with road networks before practical use, which introduces additional errors and computational overhead. In contrast, map-matched trajectory recovery integrates the road network into the model and directly recovers the trajectory on the road network. MTrajRec [26] represents each point by a road segment and moving rate, using a multi-task seq2seq framework. RNTrajRec [8] enhances this by modeling spatial-temporal relations with a transformer-based architecture. LightTR [20] introduces a lightweight recovery framework using federated learning. MM-STGED [38] further captures both local and global semantic patterns via graph modeling. While these methods demonstrate promising results, their performance is limited by the scarcity of large-scale dense trajectory data and poor generalization across different sampling intervals, as discussed in Section 1.

**Cross-domain Application of PLM** has received significant attention recently. In the field of time series analysis, Time-LLM [15] reprograms time series data with natural language prompts to harness the capabilities of PLM in handling time series effectively. TEMPO [3] leverages PLM for time series forecasting by employing interpretable prompt tuning to identify similar patterns in time series data. Similarly, TEST [29] introduces soft prompts to enhance PLM's understanding of time series embeddings. In the field of computer vision, VisionLLM [35] employs PLM as a versatile decoder and has demonstrated promising performance in diverse visual tasks. MvNet [23] integrates frozen PLM and multi-view vision prompting to efficiently encode three-dimensional data. In recommendation systems, GenRec [14] utilizes specialized prompts and extensive knowledge within PLM to provide accurate recommendations to users. Although PLMs have demonstrated effectiveness across various domains, they cannot be directly applied to trajectory learning. Trajectory data have unique spatiotemporal characteristics that require specialized modeling approaches.

## 3 Preliminaries

**Trajectory.** A trajectory is defined as a series of timestamped locations, denoted as $\mathcal{T} = \langle p_1, \cdots, p_{|\mathcal{T}|} \rangle$ where $p_i = (lat_i, lng_i, t_i)$ represents the latitude and longitude coordinates of an object at the time $t_i, i \in \{1, \cdots, |\mathcal{T}|\}$. $|\mathcal{T}|$ is the length of the trajectory. The sampling interval of $\mathcal{T}$ is defined as $t_i - t_{i-1}$, for $i \geq 2$.

**Road Network.** A road network is modeled as a directed graph $\mathcal{G} = (\mathcal{V}, \mathcal{E})$, where $\mathcal{V}$ is the set of nodes, and $\mathcal{E}$ is the set of edges. Each node $v \in \mathcal{V}$ represents an intersection and is associated with geographic coordinates, including latitude and longitude. Each edge $e \in \mathcal{E}$ corresponds to a road segment connecting two intersections, defined by its start node $e.\text{start} \in \mathcal{V}$ and end node $e.\text{end} \in \mathcal{V}$.

**Map-matched Trajectory.** Using a map-matching algorithm, a trajectory $\mathcal{T}$ can be projected onto the road network to obtain a map-matched trajectory $\mathcal{T}_m$. This ensures that each point in $\mathcal{T}_m$ aligns accurately with a particular road. A map-matched trajectory is denoted as $\mathcal{T}_m = \langle q_1, \cdots, q_{|\mathcal{T}_m|} \rangle$, where each point $q_j = (e_j, r_j, t_j)$ represents the vehicle's position at time $t_j$. Here, $e_j \in \mathcal{E}$ is the matched road segment, and $r_j$ is the moving ratio, representing the proportion of distance traveled along road segment $e_j$ relative to its total length.

**Map-matched Trajectory Recovery.** Given a sparse trajectory $\mathcal{T}_s = \langle p_1, \cdots, p_{|\mathcal{T}_s|} \rangle$ with a sampling interval of $\mu$, the goal of map-matched trajectory recovery is to reconstruct the dense map-matched trajectory $\mathcal{T}_m = \langle q_1, \cdots, q_{|\mathcal{T}_m|} \rangle$ with a sampling interval of $\epsilon$. Note that the sampling interval $\mu > \epsilon$.

## 4 Methodology

In this paper, we present both scalable and generalizable trajectory recovery model, **PLMTrajRec**, by fine-tuning a PLM that is pre-trained on a large-scale corpus with limited dense trajectory data. The overall framework of PLMTrajRec as shown in Figure 2, comprises three main components: dual trajectory prompts, an interval-aware trajectory embedder, and a PLM-based trajectory encoder. The **dual trajectory prompts** provide essential prior information through two key components. First, the interval and feature-guided (IF-guided) explicit trajectory prompt incorporates the sampling interval of sparse trajectories and their movement features into the PLM, helping the model capture

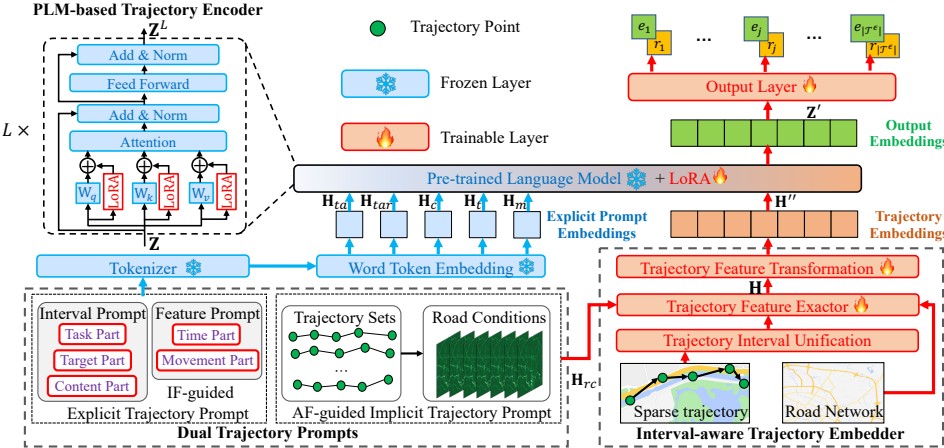

Figure 2: The framework of PLMTrajRec, consists of Dual Trajectory Prompts, Interval-aware Trajectory Embedder, and PLM-based Trajectory Encoder.

trajectory characteristics. Second, the area flow-guided (AF-guided) implicit trajectory prompt encodes road conditions, offering valuable context for recovering missing points. The **interval-aware trajectory embedder** normalizes trajectories with varying sampling intervals $\mu$ into a unified interval $\epsilon$, efficiently handling diverse spatiotemporal correlations and enhancing model generalization. Each trajectory point, both observed and missing, is then embedded into a format suitable for the PLM. The **PLM-based trajectory encoder** leverages a pre-trained BERT model trained on a large-scale corpus to capture bidirectional context. Most parameters are frozen to retain general knowledge, while a multi-head attention layer remains trainable to learn trajectory-specific patterns. Finally, the model predicts the road segment $e$ and movement ratio $r$ for the recovered map-matched trajectory.

## 4.1 Dual Trajectory Prompts

To enable PLM to recover trajectories with different intervals and effectively model road conditions for missing trajectory points, we introduce dual trajectory prompts, consisting of an IF-guided explicit trajectory prompt and an AF-guided implicit trajectory prompt.

### 4.1.1 Interval and Feature-guided (IF-guided) Explicit Trajectory Prompt

The IF-guided explicit trajectory prompt provides a structured textual description of the sampling intervals and movement features of sparse trajectories, enabling PLMTrajRec to identify varying sampling intervals and capture essential trajectory characteristics.

As shown in Figure 2, the prompts regarding sampling intervals consist of three components: the **<Task Part>** informs the PLM about the overall task to be performed, the **<Target Part>** defines the required output format, and the **<Content Part>** specifies the sampling intervals, guiding the PLM in effectively analyzing the trajectories. The prompts for movement features include two components: the **<Time Part>** provides the trajectory's specific start and end times, helping the PLM understand the duration and potential time patterns, such as morning or evening peaks. The **<Move Part>** supports the PLM in inferring the trajectory's movement patterns. We provide a detailed example of the IF-guided explicit trajectory prompt in Appendix E. After obtaining the prompt for each part, we use PLM's tokenizer and token embedding to convert text into embeddings and concatenate them to form the overall IF-guided explicit trajectory prompt embedding $\mathbf{H}^e$.

### 4.1.2 Area Flow-guided (AF-guided) Implicit Trajectory Prompt

Road conditions reflect both the surrounding environment and object movement, providing valuable information for trajectory recovery. For example, vehicles typically slow down in congested areas and accelerate in smoother traffic. However, due to the complexity and variability of real-world road conditions, describing them directly in natural language is challenging. Therefore, we represent these conditions as implicit trajectory prompts.

We first calculate the average road conditions across all areas and time intervals, then extract the relevant information for each trajectory point. Specifically, we divide the target area into a spatial grid of $I \times J$ cells and split the day into $T$ intervals. For each cell, we count the total number of passing vehicles, forming a regional flow matrix $\mathbf{RC} \in \mathbb{R}^{I \times J \times T}$, where each entry denotes the traffic volume at region $(i, j)$ and time $t$. To capture spatiotemporal patterns, we apply a 2D convolution over the spatial dimensions and a 1D convolution over the temporal dimension. This produces a feature representation of road conditions $\mathbf{H}_{rc} \in \mathbb{R}^{I \times J \times T \times F}$, where $F$ is the number of output features.

For observed points, we can directly extract their road condition features using their coordinates and timestamps. For missing points, however, extracting their road conditions is challenging since location information is unavailable. To address this, we design a road condition passing mechanism inspired by message passing [10], which estimates the road condition of missing points using nearby known information. The detailed implementation is presented in Section 4.2.2.

## 4.2 Interval-aware Trajectory Embedder

To enable our model to handle sparse trajectories with varying sampling intervals while effectively capturing spatiotemporal correlations, we propose an interval-aware trajectory embedder.

### 4.2.1 Trajectory Interval Unification

Sparse trajectories often exhibit mobility patterns at varying granularities, with sampling intervals ranging from seconds to minutes, introducing different spatiotemporal correlations between trajectory points. To address this, we normalize all input trajectories to match the sampling interval of the target trajectory. Specifically, we insert a placeholder token '[m]' to indicate missing points, resulting in a preprocessed sparse trajectory $\mathcal{T}^\epsilon$ of fixed interval $\epsilon$, where the length is given by $\frac{p_{|\mathcal{T}_s|}.t - p_1.t}{\epsilon} + 1$. Although the exact locations of '[m]' are unknown, their timestamps remain computable. We give an example about $\mathcal{T}^\epsilon$ in Appendix G.

### 4.2.2 Trajectory Feature Extractor

Given the preprocessed sparse trajectory $\mathcal{T}^\epsilon$, there are two cases for extracting trajectory point features: **Case 1**: The trajectory point $s \in \mathcal{T}^\epsilon$ is observed, i.e., $\exists k \in \{1, \cdots, |\mathcal{T}_s|\}$ such that $p_k.t = s.t$. **Case 2**: The location of trajectory point $s \in \mathcal{T}^\epsilon$ is missing, i.e., $s = [m]$, where its timestamp $s.t$ is known.

For Case 1, we incorporate both the continuous GPS coordinates of trajectory point $s$ and its local road network context to extract the spatial characteristics. First, we encode the latitude and longitude of $s$ using Learnable Fourier Features (LFF)[31, 18], which project continuous spatial inputs into a $F$-dimensional representation with a feature mapping function $\Phi(x) = W_\Phi[\cos(xW_r)||\sin(xW_r)]$, where $x \in \{s.lat, s.lon\}$. Using LFF to encode latitude and longitude features has two advantages: (1) The differences in coordinates between consecutive trajectory points are often minimal. $\Phi(\cdot)$ effectively captures these subtle positional shifts by sine and cosine functions, enhancing spatial sensitivity. (2) The relative information $x - y$ between points $x$ and $y$ can be captured through multiplication operations, which is important for understanding movement, such as distance. We provide detailed insight and proof of LFF in Appendix F.

Second, considering that vehicles move within the road network, the relationship between a trajectory point and its surrounding road segments is critical. To measure the relationship between a trajectory point $s$ and a road segment $l$, we define a function $f(d_{s,l}) = \begin{cases} e^{-(\frac{d_{s,l}}{\kappa})^2} & \text{if } d_{s,l} < \varphi_{dist}, \\ 0 & \text{otherwise}, \end{cases}$ based on their shortest distance, where $d_{s,l}$ is the shortest distance between $s$ and road $l$, $\kappa$ is a hyperparameter, and $\varphi_{dist}$ is a distance threshold. Then we obtain the road network representation $\mathbf{h}_s^{road}$ by weighting the road segment embedding (a randomly initialized vector). Finally, the complete representation of a trajectory point $s$ is obtained as:

$$\mathbf{h}_s = W_1[(\Phi(s.lat) + \Phi(s.lon)) \,||\, \mathbf{h}_s^{\text{road}}] + b_1, \tag{1}$$

where $W_1 \in \mathbb{R}^{F \times 2F}$ and $b_1 \in \mathbb{R}^F$ are learnable parameters, and $||$ is the concatenation operation.

For Case 2, where the location of point $s$ is unknown, we represent its features using road conditions, as described in Section 4.1.2. Since road conditions at a specific point are influenced by surrounding

conditions that propagate along both temporal and spatial dimensions, we propose a road condition passing mechanism, inspired by the message passing mechanism [10], to infer the road conditions at the missing location.

**Road Condition Passing Mechanism.** First, for the missing point $s$, we identify the nearest observed forward and backward trajectory points, $s_f$ and $s_b$, respectively. We give an example about $s_f$ and $s_b$ in Appendix G. Then, we retrieve the road conditions of $s_f$ and $s_b$ as follows:

$$\mathbf{h}_{rc}^f = \mathbf{H}_{rc}[\pi_{lat}(s_f.lat), \pi_{lng}(s_f.lng), \pi_t(s_f.t)], \qquad \mathbf{h}_{rc}^b = \mathbf{H}_{rc}[\pi_{lat}(s_b.lat), \pi_{lng}(s_b.lng), \pi_t(s_b.t)], \quad (2)$$

where $\pi_{lat}$, $\pi_{lng}$, and $\pi_t$ are index functions mapping latitude, longitude, and time to their corresponding indices. Next, we calculate the time intervals between point $s$ and points $s_f$ and $s_b$, denoted as $\triangle t_f = s.t - s_f.t$ and $\triangle t_b = s_b.t - s.t$. The road condition of point $s$ is then computed as:

$$\mathbf{h}_{rc}^s = \frac{e^{-\triangle t_f}\mathbf{h}_{rc}^f + e^{-\triangle t_b}\mathbf{h}_{rc}^b}{e^{-\triangle t_f} + e^{-\triangle t_b}} \tag{3}$$

In addition, we encode the time intervals $\triangle t_f$ and $\triangle t_b$ to quantify the relative position of $s$. The final representation of $s$ is $\mathbf{h}_s = W_2[\mathbf{m} \,||\, \text{FC}(\triangle t_f || \triangle t_b) \,||\, \mathbf{h}_{rc}^s] + b_2$, where $\mathbf{m} \in \mathbb{R}^F$ is a learnable vector representing the missing location, and $\text{FC}(\cdot) : \mathbb{R}^2 \to \mathbb{R}^F$ is the fully connected layer. Finally, the overall trajectory representation is $\mathbf{H} \in \mathbb{R}^{|\mathcal{T}^\epsilon| \times F}$.

### 4.2.3 Trajectory Feature Transformation

To enhance the model's ability to comprehend trajectory features, we introduce $K$ reference tokens $\mathbf{E}_w \in \mathbb{R}^{K \times F}$, inspired by [16, 11], to bridge the connection between PLM and the trajectory. These reference tokens are designed to capture the global semantics of the trajectory.

Given the trajectory embedding $\mathbf{H}$, we first apply a 1D CNN to aggregate neighboring information and capture local movement patterns, *i.e.* $\mathbf{H}' = \text{Conv1d}(\mathbf{H})$. Next, we compute cross-attention between the trajectory embedding $\mathbf{H}'$ and the reference tokens $\mathbf{E}_w$ to capture the global semantics $\mathbf{H}''$, where $\mathbf{H}'$ acts as the query, and $\mathbf{E}_w$ serves as both the key and value. Then, we concatenate $\mathbf{H}''$ with the explicit trajectory prompt embedding $\mathbf{H}^e$ to form the final trajectory representation $\mathbf{Z}$. Finally, we incorporate the Transformer positional embedding into each element of $\mathbf{Z}$ and feed this enhanced representation into the PLM encoder to encode the trajectory.

## 4.3 PLM-based Trajectory Encoder

We utilize pre-trained BERT as our foundational PLM, as its encoder-only structure is well-suited for reconstruction tasks due to its effective use of bi-directional contextual information from trajectories [29]. To optimize PLM for trajectory recovery, we implement the Low-Rank Adaptation (LoRA) algorithm [13] for fine-tuning. After obtaining the output embeddings from the PLM, we discard the IF-guided explicit trajectory prompt portion and extract trajectory embeddings $\mathbf{Z}' \in \mathbb{R}^{|\mathcal{T}^\epsilon| \times d}$. For the $j$-th element of $\mathbf{Z}'$, we apply the softmax function to calculate the probability of each road segment and determine the predicted road segment $e_j$ via the argmax operation. Additionally, we employ a Multi-Layer Perceptron (MLP) with a Sigmoid activation function to predict the moving ratio $r_j$.

## 4.4 Joint Training Strategy

To enable PLMTrajRec to effectively recover sparse trajectories across varying sampling intervals, we implement a comprehensive joint training strategy. Specifically, for each dense trajectory $\mathcal{T}$, we generate $M$ sparsified variants by resampling at different intervals, yielding $M$ distinct sparse datasets $\mathbb{T}_1, \ldots, \mathbb{T}_M$. These datasets are merged into a unified training dataset $\mathbb{T}_{\text{all}} = \bigcup_{i=1}^M \mathbb{T}_i$, enabling the model to achieve balanced performance across different sampling rates. Subsequently, we fine-tune the model on each individual sparse trajectory dataset to further enhance trajectory recovery performance for specific sampling intervals.

We employ multi-task learning to simultaneously optimize road segment recovery and moving ratio recovery. For road segment recovery, we utilize the cross-entropy loss function, while for moving ratio recovery, we employ the mean squared error loss function. These two objectives are balanced using a weighting factor $\lambda$ to ensure optimal performance on both tasks.

# 5 Experiments

In this section, we present comprehensive experiments to evaluate the effectiveness, scalability, and generalization capabilities of PLMTrajRec.

**Datasets.** We evaluate our model on four public trajectory datasets from Chengdu[3], China, and Porto[4], Portugal. For each city, we use road networks and trajectories at two different scales, named Chengdu-Small, Chengdu-Large, Porto-Small, and Porto-Large. All trajectories are standardized to a 15-second sampling interval. We remove trajectories with travel times less than 5 minutes or exceeding 1 hour, as well as outliers. The road network data is sourced from OpenStreetMap[5]. We apply a map-matching algorithm [22] to align trajectories with the road network and obtain ground truth road segments and movement ratios. Table 6 summarizes the statistics of our datasets.

**Baselines.** To evaluate the effectiveness of our model, we compare PLMTrajRec with 12 baseline methods. These include five free space trajectory recovery models: **HMM [22] + ShortestPath**, **Linear [12] + HMM [22]**, **MPR [9] + HMM [22]**, **DHTR [32] + HMM [22]**, and **AttnMove [39] + Rule**, and seven map-matched trajectory recovery models: **MTrajRec [26]**, **T2vec [17] + Decoder**, **T3s [42] + Decoder**, **TERI [7] + Decoder**, **TrajBERT [27] + Decoder**, **RNTrajRec [8]**, and **MM-STGED [38]**. The details of each baseline are provided in Appendix C.

**Evaluation Metrics.** Following existing works [26, 8, 38], we use five common metrics to evaluate the effectiveness of our model. For road segment recovery, we adopt **Accuracy (Acc)**, **Recall**, and **Precision (Prec)**. To assess the recovered GPS coordinates, we employ **Mean Absolute Error (MAE)** and **Root Mean Square Error (RMSE)**. The details of each metric are provided in Appendix D.1.

**Settings.** During training, we use sparse trajectories with three different sampling intervals $\mu$ to train PLMTrajRec, where $\mu = 4$ minutes, 2 minutes, and 1 minute, respectively, and recover dense trajectories with sampling interval $\epsilon = 15$ seconds. After training, we evaluate the model's performance across these intervals to test its generalization ability. To further improve accuracy, we fine-tune the model on each specific interval, enabling more precise trajectory recovery. We implement this fine-tuning using LoRA specifically for the attention components of the PLM. Detailed experimental settings are provided in Appendix D.2.

## 5.1 Experimental Results

The comparison of the results is shown in Table 1. We observe that as the sampling interval increases, trajectory recovery becomes more challenging, resulting in decreased performance across all models. Compared to baselines, our model achieves superior performance across all metrics, with an average improvement of 16.51% and 9.35% on the Chengdu-Small and Porto-Small datasets. Notably, our model demonstrates substantial gains in RMSE metrics, with an average reduction of 351.8 meters in Chengdu-Small and 119.2 meters in Porto-Small. These results indicate PLMTrajRec generalizes effectively, enabling accurate trajectory recovery across varying sampling intervals. By fine-tuning at the specific sampling interval, **PLMTrajRec + FT**, the performance further improves 1.49%, 0.13%, 1.16%, and 0.80% on the four datasets. These outstanding results can be attributed to the PLM's strong generalization capability in processing sparse trajectories with varying sampling intervals and the efficient extraction of spatiotemporal correlations by the interval-aware trajectory embedder.

## 5.2 Scalability Analysis

To evaluate the effectiveness of PLMTrajRec in trajectory recovery when dense trajectory data is limited, we conduct scalability experiments. Specifically, we train the model using subsets of the training set at 20%, 40%, 60%, 80%, and 100%, and evaluate performance on the test set. The results are presented in Table 2. As the size of the training set increases, the performance improves across all baseline models. Notably, with only 20% of the training data, PLMTrajRec already surpasses most baselines, trailing only MM-STGED. With 40% of the training data, PLMTrajRec outperforms all other models, demonstrating its strong scalability. This advantage underscores the practicality of PLMTrajRec in limited data scenarios.

---

[3] https://outreach.didichuxing.com/
[4] https://www.kaggle.com/competitions/pkdd-15-predict-taxi-service-trajectory-i
[5] http://www.openstreetmap.org/

Table 1: Performance comparison on Chengdu-Small and Porto-Small datasets with sampling intervals at 4 minutes, 2 minutes, and 1 minute, respectively. Red denotes the best result, and blue denotes the second-best result. ↓ means lower is better. ↑ means higher is better.

| Dataset | Methods | Acc(%) ↑ | Recall(%) ↑ | Prec(%) ↑ | MAE ↓ | RMSE ↓ |
|---|---|---|---|---|---|---|
| | | | | 4 minutes / 2 minutes / 1 minute | | |
| Chengdu-Small | HMM+ShortestPath | 26.85 / 33.85 / 35.92 | 28.64 / 47.89 / 67.92 | 29.55 / 48.31 / 60.16 | 939.3 / 754.1 / 529.7 | 1047.7 / 826.2 / 638.2 |
| | Linear+HMM | 26.42 / 43.78 / 68.59 | 30.45 / 45.35 / 65.66 | 36.15 / 48.77 / 66.67 | 974.5 / 816.9 / 707.4 | 1145.4 / 1054.7 / 1005.2 |
| | MPR+HMM | 36.93 / 49.88 / 62.25 | 38.62 / 54.62 / 62.67 | 44.53 / 50.94 / 60.53 | 821.9 / 474.8 / 418.8 | 914.1 / 899.0 / 659.0 |
| | DHTR+HMM | 41.48 / 47.17 / 51.09 | 57.34 / 60.16 / 63.40 | 50.48 / 51.73 / 50.14 | 673.6 / 662.0 / 584.7 | 911.3 / 912.2 / 750.4 |
| | AttnMove+Rule | 63.43 / 71.98 / 79.60 | 73.97 / 77.42 / 81.55 | 78.72 / 80.67 / 82.75 | 358.2 / 291.1 / 194.4 | 916.7 / 764.8 / 752.6 |
| | MTrajRec | 65.79 / 74.52 / 81.12 | 75.14 / 78.25 / 81.73 | 78.42 / 81.09 / 83.75 | 315.1 / 254.5 / 187.1 | 904.4 / 885.7 / 718.4 |
| | T3s+Decoder | 65.60 / 74.62 / 80.90 | 75.26 / 78.95 / 82.78 | 78.14 / 81.79 / 83.15 | 318.2 / 242.2 / 187.1 | 926.3 / 857.5 / 713.0 |
| | T2vec+Decoder | 66.51 / 75.69 / 81.69 | 75.68 / 78.86 / 81.90 | 78.27 / 81.68 / 83.88 | 307.5 / 231.6 / 185.6 | 915.2 / 783.6 / 714.1 |
| | TERI+Decoder | 66.42 / 75.32 / 81.25 | 75.59 / 78.74 / 81.89 | 78.36 / 81.38 / 83.92 | 309.2 / 239.0 / 186.9 | 903.8 / 823.7 / 710.4 |
| | TrajBERT+Decoder | 66.09 / 75.20 / 81.38 | 75.38 / 78.69 / 81.72 | 78.59 / 81.53 / 83.65 | 310.7 / 235.0 / 183.2 | 911.4 / 813.7 / 710.7 |
| | RNTrajRec | 67.66 / 75.80 / 81.88 | 75.59 / 79.35 / 82.09 | 79.97 / 81.86 / 84.84 | 306.1 / 218.5 / 177.9 | 886.0 / 757.0 / 702.5 |
| | MM-STGED | 70.64 / 78.14 / 84.26 | 76.04 / 80.06 / 84.15 | 81.63 / 83.58 / 85.92 | 266.2 / 197.2 / 154.0 | 829.7 / 696.0 / 633.5 |
| | **PLMTrajRec** | 74.12 / 81.76 / 87.17 | 79.63 / 84.31 / 87.99 | 86.46 / 88.38 / 90.52 | 262.8 / 187.4 / 141.4 | 483.0 / 366.2 / 290.8 |
| | **PLMTrajRec+FT** | 74.58 / 82.29 / 88.15 | 80.09 / 84.59 / 88.82 | 86.63 / 88.72 / 91.04 | 253.2 / 181.2 / 138.6 | 465.7 / 349.0 / 289.1 |
| Porto-Small | HMM+ShortestPath | 20.19 / 27.30 / 34.75 | 26.22 / 40.05 / 48.46 | 33.51 / 46.00 / 48.67 | 886.9 / 647.3 / 527.2 | 941.5 / 747.0 / 659.3 |
| | Linear+HMM | 32.23 / 49.35 / 66.17 | 36.09 / 50.45 / 64.72 | 49.80 / 63.87 / 75.22 | 974.3 / 408.7 / 368.3 | 637.3 / 609.8 / 571.1 |
| | MPR+HMM | 32.22 / 49.76 / 66.27 | 38.67 / 52.97 / 65.66 | 48.07 / 62.86 / 74.51 | 534.0 / 409.8 / 402.6 | 700.2 / 610.9 / 628.2 |
| | DHTR+HMM | 32.02 / 43.76 / 52.98 | 58.29 / 65.25 / 69.60 | 45.61 / 52.10 / 57.17 | 456.7 / 385.3 / 420.4 | 627.1 / 578.0 / 625.8 |
| | AttnMove+Rule | 49.31 / 61.39 / 72.07 | 48.62 / 60.90 / 69.59 | 78.03 / 82.98 / 80.41 | 310.0 / 213.4 / 156.6 | 621.3 / 468.5 / 360.7 |
| | MTrajRec | 52.36 / 61.65 / 71.65 | 60.39 / 65.65 / 70.92 | 77.28 / 78.99 / 80.35 | 266.1 / 179.9 / 114.9 | 590.1 / 451.5 / 332.3 |
| | T3s+Decoder | 52.24 / 61.75 / 71.78 | 60.24 / 65.53 / 71.61 | 77.80 / 79.14 / 80.16 | 270.4 / 181.3 / 110.1 | 594.9 / 461.2 / 328.0 |
| | T2vec+Decoder | 53.13 / 62.24 / 71.86 | 60.27 / 65.77 / 71.10 | 77.62 / 78.97 / 80.48 | 256.3 / 173.9 / 114.2 | 571.0 / 438.0 / 334.7 |
| | TERI+Decoder | 53.59 / 62.14 / 71.53 | 60.02 / 65.39 / 71.38 | 78.26 / 79.18 / 80.29 | 253.9 / 178.3 / 113.2 | 558.3 / 443.9 / 325.9 |
| | TrajBERT+Decoder | 52.98 / 62.34 / 71.63 | 60.12 / 65.66 / 71.47 | 77.93 / 79.02 / 80.39 | 251.8 / 177.9 / 112.7 | 560.1 / 441.2 / 329.5 |
| | RNTrajRec | 54.59 / 63.39 / 72.31 | 60.42 / 65.84 / 71.88 | 79.20 / 79.25 / 80.57 | 248.1 / 171.3 / 110.3 | 549.1 / 433.9 / 325.7 |
| | MM-STGED | 57.30 / 65.69 / 73.16 | 59.48 / 66.15 / 72.27 | 80.21 / 80.74 / 80.81 | 222.8 / 152.5 / 108.2 | 510.4 / 400.8 / 321.9 |
| | **PLMTrajRec** | 57.61 / 66.40 / 74.42 | 59.15 / 66.52 / 72.78 | 82.19 / 82.14 / 82.82 | 200.9 / 141.9 / 95.6 | 376.9 / 294.6 / 211.7 |
| | **PLMTrajRec+FT** | 57.67 / 66.72 / 75.35 | 59.08 / 66.87 / 73.87 | 82.05 / 82.38 / 83.28 | 200.8 / 141.7 / 95.2 | 370.2 / 293.5 / 220.3 |
| Chengdu-Large | HMM+Short | 22.38 / 38.72 / 48.36 | 27.18 / 45.46 / 57.57 | 28.69 / 49.99 / 55.42 | 886.2 / 572.8 / 388.3 | 1136.0 / 846.7 / 638.6 |
| | Linear+HMM | 25.71 / 36.12 / 56.50 | 35.06 / 41.53 / 62.11 | 36.05 / 46.11 / 68.52 | 718.7 / 529.3 / 375.7 | 979.2 / 761.0 / 611.6 |
| | MPR | 31.10 / 45.70 / 58.68 | 41.57 / 52.08 / 64.32 | 47.43 / 58.30 / 70.21 | 718.7 / 529.3 / 396.8 | 946.5 / 793.3 / 673.3 |
| | DHTR+HMM | 37.45 / 52.99 / 66.05 | 50.47 / 59.71 / 71.95 | 56.35 / 64.98 / 74.23 | 582.5 / 492.3 / 248.5 | 930.9 / 776.9 / 572.1 |
| | AttnMove+HMM | 67.37 / 73.53 / 79.14 | 71.16 / 77.92 / 80.61 | 72.61 / 78.46 / 81.60 | 361.8 / 344.7 / 197.5 | 859.0 / 663.5 / 455.7 |
| | MTrajRec | 70.09 / 80.09 / 83.75 | 74.11 / 81.72 / 84.46 | 75.17 / 81.64 / 85.19 | 266.1 / 312.4 / 146.4 | 828.5 / 528.6 / 347.7 |
| | T3s+Decoder | 70.62 / 79.19 / 83.11 | 75.23 / 83.65 / 85.58 | 77.94 / 82.97 / 85.84 | 359.1 / 293.8 / 127.3 | 771.3 / 499.5 / 299.8 |
| | T2vec+Decoder | 71.49 / 80.11 / 82.35 | 76.97 / 81.57 / 84.96 | 78.38 / 82.76 / 85.84 | 301.8 / 246.2 / 135.6 | 792.8 / 461.4 / 301.8 |
| | TREI+Decoder | 71.20 / 81.81 / 83.72 | 75.04 / 83.54 / 86.48 | 78.04 / 81.11 / 87.02 | 267.6 / 205.3 / 125.6 | 783.3 / 400.8 / 278.5 |
| | TrajBERT+Decoder | 71.15 / 80.57 / 84.37 | 75.88 / 82.35 / 86.91 | 79.61 / 81.03 / 86.96 | 278.8 / 218.3 / 119.7 | 752.6 / 413.0 / 282.1 |
| | RNTrajRec | 73.62 / 80.85 / 84.21 | 76.05 / 83.02 / 86.34 | 78.90 / 82.73 / 87.75 | 261.8 / 183.3 / 121.8 | 764.8 / 389.8 / 253.0 |
| | MM-STGED | 75.51 / 83.77 / 86.21 | 79.24 / 84.02 / 89.17 | 80.13 / 85.57 / 89.22 | 240.2 / 162.9 / 102.0 | 718.9 / 337.7 / 203.9 |
| | **PLMTrajRec** | 78.97 / 85.11 / 89.84 | 82.57 / 88.72 / 92.38 | 86.13 / 89.01 / 93.11 | 222.8 / 146.9 / 85.6 | 421.6 / 273.0 / 137.5 |
| | **PLMTrajRec+FT** | 79.15 / 85.48 / 90.28 | 83.03 / 88.93 / 92.22 | 86.96 / 89.34 / 92.86 | 218.2 / 143.1 / 83.6 | 406.6 / 268.6 / 133.0 |
| Porto-Large | HMM+Short | 23.81 / 43.10 / 49.16 | 28.57 / 52.40 / 53.50 | 26.17 / 50.40 / 57.30 | 810.4 / 717.7 / 387.7 | 932.9 / 704.5 / 496.3 |
| | Linear+HMM | 28.75 / 43.37 / 57.62 | 31.35 / 51.14 / 61.90 | 34.11 / 57.91 / 63.94 | 483.5 / 634.8 / 349.6 | 961.2 / 683.7 / 462.1 |
| | MPR | 37.14 / 46.54 / 58.67 | 39.08 / 55.82 / 60.91 | 38.58 / 58.30 / 62.42 | 649.6 / 497.3 / 355.2 | 830.1 / 552.1 / 458.7 |
| | DHTR+HMM | 39.87 / 54.78 / 63.75 | 50.38 / 60.26 / 64.81 | 57.44 / 58.25 / 66.53 | 518.3 / 439.2 / 241.2 | 663.8 / 466.9 / 362.8 |
| | AttnMove+HMM | 57.25 / 63.05 / 68.22 | 60.08 / 65.76 / 70.64 | 66.47 / 66.59 / 69.77 | 332.9 / 263.3 / 174.9 | 414.5 / 376.6 / 288.0 |
| | MTrajRec | 61.87 / 66.65 / 72.63 | 62.99 / 68.68 / 72.42 | 69.44 / 69.55 / 71.63 | 259.9 / 194.4 / 128.0 | 362.1 / 320.5 / 216.5 |
| | T3s+Decoder | 63.52 / 67.39 / 71.31 | 64.92 / 70.56 / 73.67 | 72.81 / 71.06 / 74.08 | 216.0 / 172.2 / 137.0 | 375.1 / 324.9 / 225.9 |
| | T2vec+Decoder | 64.24 / 68.84 / 72.54 | 63.85 / 69.70 / 74.60 | 73.86 / 70.04 / 72.82 | 231.1 / 164.8 / 103.6 | 358.4 / 308.3 / 208.1 |
| | TREI+Decoder | 63.41 / 67.63 / 72.21 | 63.71 / 69.54 / 73.59 | 74.78 / 71.37 / 74.22 | 215.9 / 166.5 / 115.4 | 345.7 / 311.7 / 212.4 |
| | TrajBERT+Decoder | 63.59 / 68.54 / 71.59 | 62.09 / 70.75 / 74.62 | 73.58 / 70.83 / 73.32 | 191.0 / 150.7 / 91.6 | 342.2 / 293.6 / 190.2 |
| | RNTrajRec | 65.98 / 68.88 / 72.72 | 64.75 / 70.71 / 74.31 | 74.94 / 71.25 / 73.95 | 192.8 / 147.6 / 82.4 | 311.0 / 286.6 / 185.8 |
| | MM-STGED | 66.66 / 70.03 / 74.03 | 65.59 / 73.44 / 76.68 | 75.62 / 74.53 / 76.33 | 173.6 / 113.6 / 68.6 | 296.3 / 255.9 / 149.6 |
| | **PLMTrajRec** | 67.57 / 73.14 / 77.30 | 68.74 / 76.10 / 79.36 | 78.93 / 76.77 / 80.92 | 146.3 / 97.3 / 53.8 | 264.2 / 191.9 / 116.7 |
| | **PLMTrajRec+FT** | 67.87 / 73.68 / 77.75 | 69.18 / 76.40 / 79.93 | 78.87 / 77.02 / 81.05 | 142.7 / 96.0 / 52.3 | 261.3 / 188.2 / 117.3 |

Table 2: Scalability analysis. The performance comparison on the Chengdu-Small dataset when trained with different data ratios. The result of other dataset can be found in Table 7, 8, and 9. Red denotes the best result, and blue denotes the second-best result.

| Setting | Data Ratio | 20% | | 40% | | 60% | | 80% | | 100% | |
|---|---|---|---|---|---|---|---|---|---|---|---|
| | Metric | Acc(%) | RMSE | Acc(%) | RMSE | Acc(%) | RMSE | Acc(%) | RMSE | Acc(%) | RMSE |
| $\mu = 4$ minutes | MTrajRec | 60.73 | 1048.0 | 63.58 | 968.5 | 64.73 | 929.4 | 65.39 | 914.6 | 65.79 | 904.4 |
| | T3s + Decoder | 60.49 | 1098.9 | 63.32 | 968.7 | 64.23 | 961.9 | 65.04 | 942.3 | 65.60 | 926.3 |
| | T2vec + Decoder | 62.17 | 966.1 | 64.92 | 946.1 | 64.95 | 972.3 | 65.31 | 946.8 | 66.51 | 915.2 |
| | RNTrajRec | 60.80 | 998.0 | 62.85 | 931.3 | 64.19 | 909.1 | 65.46 | 835.0 | 67.66 | 886.0 |
| | MM-STGED | 64.61 | 935.0 | 67.76 | 881.8 | 68.53 | 853.7 | 69.95 | 825.3 | 70.64 | 829.7 |
| | **PLMTrajRec** | 68.30 | 585.3 | 71.57 | 515.6 | 72.74 | 512.5 | 73.56 | 488.3 | 74.12 | 483.0 |
| $\mu = 2$ minutes | MTrajRec | 69.85 | 919.8 | 72.58 | 907.4 | 73.97 | 902.3 | 74.32 | 891.3 | 74.52 | 885.7 |
| | T3s + Decoder | 68.86 | 909.5 | 72.29 | 897.2 | 73.63 | 883.8 | 74.53 | 868.8 | 74.62 | 857.5 |
| | T2vec + Decoder | 69.47 | 905.4 | 72.77 | 858.3 | 73.80 | 855.8 | 74.89 | 831.3 | 75.69 | 783.6 |
| | RNTrajRec | 68.05 | 991.0 | 70.20 | 856.9 | 71.17 | 834.4 | 72.62 | 795.3 | 75.80 | 757.0 |
| | MM-STGED | 71.65 | 865.0 | 75.32 | 773.0 | 75.49 | 752.1 | 76.94 | 747.7 | 78.14 | 696.0 |
| | **PLMTrajRec** | 76.29 | 452.8 | 79.37 | 411.5 | 80.37 | 395.4 | 81.18 | 377.4 | 81.76 | 366.2 |
| $\mu = 1$ minute | MTrajRec | 75.39 | 937.5 | 78.53 | 835.1 | 80.01 | 794.3 | 80.93 | 725.7 | 81.12 | 718.4 |
| | T3s + Decoder | 76.49 | 917.1 | 79.08 | 824.9 | 80.65 | 767.8 | 80.82 | 716.5 | 80.90 | 713.0 |
| | T2vec + Decoder | 75.89 | 845.4 | 79.09 | 746.8 | 79.41 | 752.3 | 81.21 | 742.2 | 81.69 | 714.1 |
| | RNTrajRec | 75.65 | 846.2 | 79.48 | 784.8 | 79.61 | 769.3 | 81.74 | 742.5 | 81.88 | 702.5 |
| | MM-STGED | 76.02 | 857.7 | 79.86 | 734.7 | 82.25 | 676.4 | 83.44 | 663.0 | 84.26 | 633.5 |
| | **PLMTrajRec** | 81.62 | 397.5 | 84.59 | 352.0 | 85.75 | 328.7 | 86.73 | 305.4 | 87.17 | 290.8 |

## 5.3 Zero-shot Study on Sampling Interval

We further analyze the model's generalizability when tested on sampling intervals absent during training. Specifically, PLMTrajRec is trained with sampling intervals of 1 minute and 4 minutes, then evaluated on trajectories sampled at 2-minute intervals. Table 3 presents the results, where ↓

Table 3: Zero-shot study on Chengdu-Small and Porto-Small datasets with $\mu = 2$ minutes. Red denotes the best result, and blue denotes the second-best result.

| Datasets | Chengdu-Small | | | Porto-Small | | |
|---|---|---|---|---|---|---|
| Methods | Acc(%) | MAE | RMSE | Acc(%) | MAE | RMSE |
| MTrajRec | 56.39 (↓18.13%) | 492.7 (↓48.37%) | 1046.2 (↓15.34%) | 37.51 (↓24.14%) | 322.0 (↓44.13%) | 566.8 (↓20.34%) |
| T3s + Decoder | 63.81 (↓10.81%) | 353.0 (↓31.39%) | 980.6 (↓9.68%) | 45.88 (↓15.87%) | 247.2 (↓26.66%) | 527.1 (↓12.51%) |
| T2vec + Decoder | 63.32 (↓12.37%) | 376.6 (↓38.50%) | 928.1 (↓15.57%) | 51.46 (↓10.78%) | 245.3 (↓29.10%) | 507.2 (↓13.65%) |
| RNTrajRec | 64.93 (↓10.87%) | 339.6 (↓35.70%) | 912.8 (↓17.07%) | 52.82 (↓10.57%) | 214.1 (↓19.98%) | 478.6 (↓9.34%) |
| MM-STGED | 68.27 (↓9.90%) | 291.8 (↓32.42%) | 862.6 (↓19.31%) | 55.71 (↓9.98%) | 182.4 (↓16.40%) | 450.9 (↓11.11%) |
| **PLMTrajRec** | **79.62 (↓2.14%)** | **195.3 (↓4.05%)** | **379.6 (↓3.53 %)** | **64.96 (↓1.44%)** | **147.1 (↓3.52%)** | **303.8 (↓3.03%)** |

indicates the percentage decline compared to performance when trained with 2-minute sampling intervals. While all models exhibit decreased accuracy when the target sampling interval is excluded from training data, PLMTrajRec demonstrates a notably smaller performance degradation. This robustness can be attributed to its trajectory prompts and joint training strategy, which effectively identify the sampling interval patterns in target trajectories, enhancing generalizability. In contrast, baseline models rely exclusively on trajectory point information to capture correlations, limiting their ability to generalize across different temporal dynamics between training and testing datasets.

## 5.4 Model Analysis

Table 4: Ablation study with $\mu = 2$ minutes. Red denotes the best result, and blue denotes the second-best result.

| Datasets | Chengdu-Small | | | Porto-Small | | | Chengdu-Large | | | Porto-Large | | |
|---|---|---|---|---|---|---|---|---|---|---|---|---|
| Methods | Acc(%) | MAE | RMSE | Acc(%) | MAE | RMSE | Acc(%) | MAE | RMSE | Acc(%) | MAE | RMSE |
| PLMTrajRec - Randomly initialized BERT | 74.11 | 288.4 | 831.9 | 62.06 | 179.3 | 413.9 | 79.28 | 251.5 | 462.0 | 68.29 | 169.2 | 282.3 |
| PLMTrajRec - GPT-2 | 80.23 | 273.7 | 492.7 | 64.33 | 272.0 | 450.6 | 81.63 | 217.5 | 428.6 | 70.29 | 149.3 | 266.5 |
| PLMTrajRec - Llama | 80.51 | 255.3 | 473.5 | 64.69 | 235.1 | 438.3 | 82.24 | 204.1 | 412.4 | 70.58 | 136.9 | 248.1 |
| w/o IF-guided explicit trajectory prompt | 81.04 | 205.8 | 394.2 | 66.13 | 157.9 | 314.6 | 83.82 | 167.0 | 322.7 | 72.24 | 125.0 | 236.0 |
| w/o AF-guided implicit trajectory prompt | 81.43 | 191.4 | 371.1 | 66.18 | 149.6 | 298.1 | 84.23 | 173.9 | 304.2 | 72.10 | 128.3 | 241.1 |
| w/o Dual trajectory prompts | 80.70 | 225.9 | 407.4 | 65.52 | 171.0 | 339.4 | 83.20 | 185.3 | 348.2 | 71.72 | 131.0 | 249.1 |
| w/o Reference tokens | 80.83 | 236.6 | 437.2 | 65.89 | 165.4 | 327.7 | 82.71 | 206.3 | 386.6 | 71.26 | 131.2 | 251.3 |
| **PLMTrajRec** | **81.76** | **187.4** | **366.2** | **66.40** | **141.9** | **294.6** | **85.11** | **146.9** | **273.0** | **73.14** | **97.3** | **191.9** |

**Ablation Study.** As shown in Table 4, removing dual trajectory prompts results in a 12.80% degradation in average performance across the four datasets, validating the importance of capturing both task-specific information and road conditions for missing trajectory points. Ablating reference tokens causes performance declines ranging from 8.27% to 20.19%, as the PLM cannot effectively interpret raw trajectory data without these references. Experiments with randomly initialized BERT in PLMTrajRec yield substantially poorer results, demonstrating the effectiveness of pre-trained language models for trajectory recovery task. Similarly, decoder-only PLM frameworks such as GPT-2 and Llama show decreased performance, indicating that the ability to capture bidirectional contextual information is essential for accurate trajectory recovery.

Table 5: Efficiency analysis on Chengdu-Small and Porto-Small datasets with $\mu = 2$ minutes. The ratio after the trainable parameters denotes the percentage of trainable parameters to the total parameters.

| Dataset | Chengdu-Small | | | | Porto-Small | | | |
|---|---|---|---|---|---|---|---|---|
| Methods | # Param. | # Trainable Param. (ratio) | Train time (min/epoch) | Inference time (min) | # Param. | # Trainable Param. (ratio) | Train time (min/epoch) | Inference time (min) |
| DHTR | 4.95M | 4.95M (100%) | 7.37 | 1.29 | 4.62M | 4.62M (100%) | 10.52 | 3.07 |
| MTrajRec | 5.03M | 5.03M (100%) | 9.58 | 1.86 | 4.85M | 4.85M (100%) | 18.68 | 5.41 |
| RNTrajRec | 11.36M | 11.36M (100%) | 10.73 | 2.23 | 10.85M | 10.85M (100%) | 20.57 | 6.83 |
| MM-STGED | 9.24M | 9.24M (100%) | 12.35 | 5.77 | 8.88M | 8.88M (100%) | 28.27 | 16.39 |
| **PLMTrajRec** | 45.05M | 16.28M (35.15%) | 24.18 | 2.56 | 44.76M | 16.00M (35.74%) | 54.14 | 10.38 |

**Efficiency Analysis.** As shown in Table 5, incorporating a PLM increases the model's parameter count as well as training and inference time. However, by deploying LoRA, we significantly reduce the number of trainable parameters, with only 35.15% and 35.74% of the total parameters requiring training. Since the introduction of PLM substantially improves trajectory recovery performance, this approach achieves an effective trade-off between performance efficacy and computational efficiency.

## 5.5 Hyperparameter Study

To explore the impact of hyperparameters on model performance, we conduct hyperparameter analysis with the sparse trajectory sampling interval of 2 minutes.

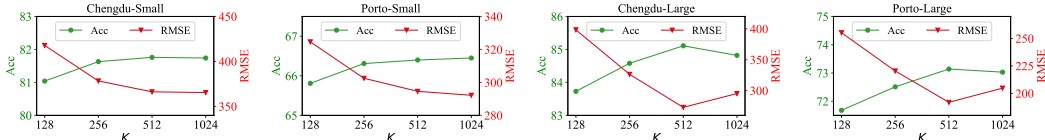

Figure 3: Hyperparameter analysis of the number of reference tokens $K$ of trajectory feature transformation.

**The Number of Reference Tokens** $K$ We set the parameter $K$ from the set $\{128, 256, 512, 1024\}$ to explore its impact on trajectory recovery. The experimental results are shown in Figure 3. As $K$ increases, the effectiveness of trajectory recovery also improves. This suggests that having a larger token space aids in accurately representing trajectory features. When $K$ is larger than 512, the performance improvement of PLMTrajRec becomes marginal, while the computational cost will increase. To balance the performance and efficiency of the model, we set $K$ to 512.

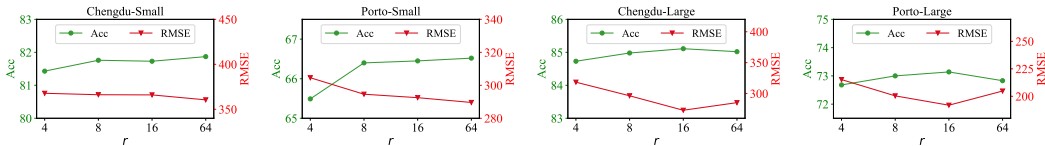

Figure 4: Hyperparameter analysis of the rank $r$ of LoRA.

**The Rank** $r$ **of LoRA** As shown in Figure 4, we set the range of $r \in \{4, 8, 16, 64\}$ to explore its impact. It is observed that the model performs well when $r = 4$. As $r$ increases, the accuracy will continue to increase, but not obvious. This suggests that PLM encapsulates substantial domain expertise through training on extensive corpora, rendering it adaptable to trajectory recovery tasks with minor adjustments. Yet as $r$ increases, the number of parameters also increases, making the model training require more memory. Thus, we set $r = 8$ to balance the performance and resources.

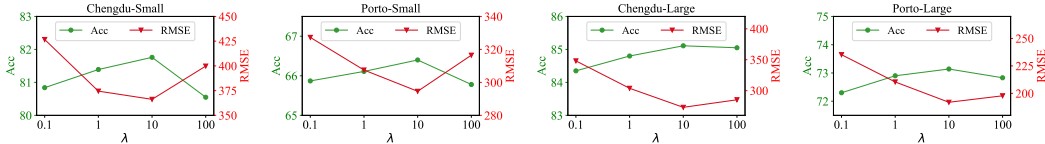

Figure 5: Hyperparameter analysis of the weight $\lambda$ in the loss function.

**The Loss Weight** $\lambda$ As shown in Figure 5, as the loss function weight $\lambda$ increases, the model performance first improves and then decreases. This is because a smaller $\lambda$ will make the model focus on road segment recovery, while a larger $\lambda$ will focus on moving rate recovery. To balance these two tasks, we set the value of $\lambda$ to 10.

## 6 Conclusion

In this paper, we investigate the problem of trajectory recovery with limited-scale training datasets and propose a novel trajectory recovery model named PLMTrajRec. By leveraging pre-trained language models, PLMTrajRec can effectively recover trajectories even with limited dense trajectory datasets, thereby demonstrating strong scalability. The model incorporates both an interval and feature-guided explicit trajectory prompt and an interval-aware trajectory embedder, enabling it to effectively generalize across sampling intervals. Additionally, we introduce an area flow-guided implicit trajectory prompt to gather traffic flows in each region, and propose a road condition passing mechanism to infer missing-point conditions from nearby observations. Experimental results on four datasets with three sampling intervals validate the effectiveness, scalability, and generalizability of the proposed model. We discuss the limitations and broader impacts of PLMTrajRec in Appendix A.

**Acknowledgment.** This work was supported by the National Natural Science Foundation of China (No. 62372031), A*STAR RIE2025 Manufacturing, Trade and Connectivity (MTC) Programmatic Fund (M24N6b0043), and ECNU Multifunctional Platform for Innovation (001).

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

# A    Limitations and Broader Impacts

## A.1    Limitations

Due to the varying number of road segments across different datasets, some modules are technically not transferable across datasets, such as the road segment embeddings and the road segment ID prediction based on multi-classification. Consequently, our model trained on one dataset struggles to generalize to others. Future work will focus on developing a universal road segment embedding to enable cross-dataset adaptation and enhance model versatility.

## A.2    Broader Impacts

Our proposed model demonstrates the effectiveness of applying PLM to trajectory data. The inherent scalability and generalization capabilities of PLM can be effectively leveraged for trajectory modeling, enabling the model to achieve comparable performance using only 20% of the training data and to generalize well to unseen sampling intervals. Furthermore, for the trajectory recovery task, encoder-only PLM architectures outperform decoder-only, as they are better suited to capture bidirectional information within trajectories.

# B    Dataset

Following [38], for -Small datasets, we focus on the road network only on the commonly used roads. Specifically, for the Chengdu-Small dataset, we select four types of roads: Primary, Secondary, Trunk, and Tertiary. For the Porto-Small, we select Primary, Secondary, Motorway, and Tertiary. We employ map matching on the selected road network to obtain the ground-truth road segment ID and moving ratio of the trajectory, and trajectories that cannot be matched are discarded. In addition, to validate our approach on a larger scale, we utilize all roads within the regions and perform map matching again, and construct a larger dataset, termed as Chengdu-Large and Porto-Large. The statistics of these datasets are summarized in Table 6.

Table 6: Statistics Description of Dataset.

| Types | Chengdu-Small | Porto-Small | Chengdu-Large | Porto-Large |
|---|---|---|---|---|
| #Sampling Intervals | 15s | 15s | 15s | 15s |
| #Trajectories | 118,354 | 322,079 | 1,563,292 | 836,391 |
| #Road Segments | 2504 | 2224 | 12564 | 15707 |
| Latitude range | (30.655, 30.727) | (41.142, 41.174) | (30.600, 30.730) | (41.100, 41.190) |
| Longitude range | (104.043, 104.129) | (-8.652, -8.578) | (104.011, 104.150) | (-8.675, -8.550) |

# C    Baseline Setting

We choose the following 12 methods as baselines, including five free space trajectory recovery and seven map-matched trajectory recovery.

## C.1    Free-space Trajectory Recovery

Free space trajectory recovery first recovers trajectory points and then projects the trajectory onto the road network.

- **HMM [22] + ShortestPath** first projects the sparse trajectory on the road network based on the hidden markov model (HMM), and then calculate the shortest path.

- **Linear [12] + HMM [22]** linearly interpolates missing trajectory points and then implements HMM to perform the map matching process.

- **MPR [9] + HMM [22]** first divides the area of interest into grids to identify frequently traveled routes between sparse trajectory points. Then, it assumes that vehicle movement maintains a

constant speed for trajectory recovery. Subsequently, it employs HMM to project the trajectory onto the road network.

- **DHTR [32] + HMM [22]** incorporates a sequence-to-sequence framework and Kalman filtering to recover trajectory points. Subsequently, it applies an HMM to yield a trajectory that is constrained by the map.
- **AttnMove [39] + Rule** uses attention to predict the missing road segments and uses the central location as the moving rate.

## C.2  Map-matched Trajectory Recovery

Map-matched trajectory recovery can directly recover the trajectory on the road network.

- **MTrajRec [26]** utilizes a sequence-to-sequence framework with Gated Recurrent Units (GRU) as the key component for trajectory recovery. It optimizes road segment and moving rate prediction through multi-task learning.
- **T2vec [17]** is a deep learning model for trajectory similarity learning. We use its encoder to embed the sparse trajectory.
- **T3s [42]** uses LSTM and attention mechanisms to encode sparse trajectory data effectively.
- **TERI [7]** assumes that the number of missing trajectory points is unknown and proposes a two-stage trajectory recovery framework. In the first stage, a transformer-based model predicts the number of points to be recovered, and in the second stage, the same framework is used for recovery trajectory coordinates. Here, we utilize only the second stage of TERI.
- **TrajBERT [27]** employs a transformer encoder and a forward and backward neighbor selector to learn complex mobility patterns bi-directionally from sparse trajectories.
- **RNTrajRec [8]** leverages the Transformer to capture the spatial-temporal correlation of the sparse trajectories. It also takes into account the relation between the trajectory and the road network.
- **MM-STGED [38]** models sparse trajectories from a graph perspective and recovers trajectories by capturing micro and macro semantic information.

Notably, despite T2vec, T3S, TERI, and TrajBERT employing different techniques to capture sparse trajectories' spatial-temporal dependencies, they cannot directly generate the desired format for missing points. Therefore, after these models obtain the trajectory embeddings, we append the Decoder part of MTrajRec to output the road segment and moving ratio of the trajectory point. Denoted by **T2v + Decoder**, **T3S + Decoder**, **TERI + Decoder**, and **TrajBERT + Decoder**, respectively.

# D  Implement Details

## D.1  Evaluation Metrics

We adopt five widely used metrics to evaluate the effectiveness of our model, following previous works [38, 8, 26]. For road segment recovery, we use **Accuracy (Acc)**, **Recall**, and **Precision (Prec)** to assess the alignment between the true road segments $\mathcal{E}_p = \{e_1, \cdots, e_m\}$ and the predicted road segments $\hat{\mathcal{E}}_p = \{\hat{e}_1, \cdots, \hat{e}_m\}$. A higher value in these metrics indicates a more accurate road segment recovery. The metrics are formally defined as follows:

$$\textbf{Acc} = \frac{1}{m} \sum_{i=1}^{m} \mathbb{1}\{e_i = \hat{e}_i\} \times 100\%,$$

$$\textbf{Recall} = \frac{|\mathcal{E}_p \cap \hat{\mathcal{E}}_p|}{|\mathcal{E}_p|} \times 100\%, \tag{4}$$

$$\textbf{Prec} = \frac{|\mathcal{E}_p \cap \hat{\mathcal{E}}_p|}{|\hat{\mathcal{E}}_p|} \times 100\%,$$

where $\mathbb{1}\{\cdot\}$ is the indicator function, where $e_i = \hat{e}_i$, $\mathbb{1}\{e_i = \hat{e}_i\} = 1$, elsewise $\mathbb{1}\{e_i = \hat{e}_i\} = 0$.

To evaluate the recovered GPS coordinates, we employ the **Mean Absolute Error (MAE)** and **Root Mean Square Error (RMSE)** to quantify the distance error between the true trajectory $\mathcal{T}_m = q_1, \cdots, q_{|\mathcal{T}_m|}$ and the predicted trajectory $\hat{\mathcal{T}}_m = \hat{q}_1, \cdots, \hat{q}_{|\hat{\mathcal{T}}_m|}$. The formulas for MAE and RMSE are given as follows:The formulas of MAE and RMSE are as follows:

$$
\mathbf{MAE} = \frac{1}{|\mathcal{T}_m|} \sum_{i=1}^{|\mathcal{T}_m|} |\mathrm{RN\_dist}(q_i, \hat{q}_i)|,
$$

$$
\mathbf{RMSE} = \sqrt{\frac{1}{|\mathcal{T}_m|} \sum_{i=1}^{|\mathcal{T}_m|} |\mathrm{RN\_dist}(q_i, \hat{q}_i)|^2}
$$

(5)

Here, following [26, 8, 38], $\mathrm{RN\_dist}(q, \hat{q})$ signifies the shortest distance along the road network between the trajectory points $q$ and $\hat{q}$. Both MAE and RMSE are denoted in meters. Lower values of these metrics indicate a higher level of accuracy in the recovery results.

### D.2 Setting

We split the trajectory dataset into training, validation, and testing sets in a 7:2:1 ratio. We employ the PyTorch framework to implement PLMTrajRec, with a learning rate of 1e-4 and a batch size of 64. BERT-small is selected as the foundation model for PLM with 4 transformer layers, and the number of hidden states is 512. For road condition extraction, the area of interest is divided into a 64 × 64 grid, and the time dimension is partitioned into hourly intervals. The hidden state dimension is set to $F = 512$. In function $f(d_{s,l})$ of Section 4.2.2, we set $\kappa = 15$ and $\varphi_{dist} = 50$ meters. The model is trained for 50 epochs with early stopping, using a patience of 10 epochs. We train the baselines using the parameters reported in the original paper and set the number of training epochs to 50. All experiments are conducted on NVIDIA RTX A4000 GPUs.

## E  Example of the IF-guided Explicit Trajectory Prompt

Consider a sparse trajectory $\mathcal{T} = \langle p_1, \cdots, p_N \rangle$ of $N$ trajectory points with a sampling interval of 4 minutes, starting at 8 o'clock on Saturday and ending at 9 o'clock on Saturday. Our goal is to recover it within a sampling interval of 15 seconds. Therefore, the trajectory prompts that are related to sampling intervals are:

- **Task Part**: Sparse trajectory recovery.

- **Target Part**: Output the road segment and moving ratio for each point in the trajectory.

- **Content Part**: The sparse trajectory is sampled on average {*four minutes*} and aims to recover trajectory every {*fifteen seconds*}.

The content within the placeholders {} is filled with trajectory-specific information. The movement feature-related trajectory prompts are:

- **Time Part**: The trajectory started at {*eight o'clock*} on {*Saturday*} and ended at {*nine o'clock*} on {*Saturday*}.

- **Movement Part**: Total time cost: {*sixty minutes zero seconds*}. Total space transfer distance: {*z*} kilometers.

where the elements in  are determined by the characteristics of the sparse trajectory. Here $z = \sum_{i=2}^{N} \mathrm{dist}(p_i, p_{i-1})$, where $\mathrm{dist}(\cdot, \cdot)$ is used to calculate the distance between two trajectory points.

## F  Insight about the Learnable Fourier Features

Consider two trajectory points $x$ and $y$, and the feature mapping function $\Phi(x) = W_\Phi[\cos x W_r || \sin x W_r]$ in Learnable Fourier Features. The relative information $x - y$ between

points $x$ and $y$ can be captured through multiplication operations, i.e.:

$$\Phi(x) \cdot \Phi(y) = W_\Phi[\cos xW_r || \sin xW_r] \cdot W_\Phi[\cos yW_r || \sin yW_r]$$
$$= ||W_\phi||_2 \cdot (\cos x \cdot \cos y + \sin x \cdot \sin y) \cdot ||W_r||_2 \qquad (6)$$
$$= ||W_\phi||_2 \cdot \cos(x - y) \cdot ||W_r||_2$$

In our PLMTrajRec, after feature conversion, we input the trajectory feature into a pre-language training model based on BERT. Since there are a lot of multiplication-based attention operations in the PLM, the relative information $x-y$ can be easily modeled and utilized. This relative information helps infer crucial details like the distance between trajectory points, which is important for understanding movement. For instance, a larger distance between two points may indicate a higher likelihood of vehicle acceleration.

## G  Example of the Preprocessed Sparse Trajectory and Road Condition of the Missing Trajectory Point

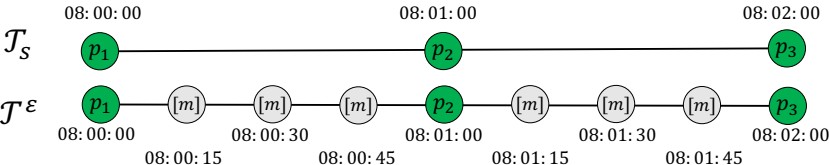

Figure 6: An example of the processed sparse trajectory.

As shown in Figure 6, the sparse trajectory $\mathcal{T}_s$ consists of three trajectory points $p_1, p_2$, and $p_3$, with a sampling interval of 1 minute. We aim to reconstruct the dense trajectory with a sampling interval of 15 seconds. Based on the timestamps of the observed trajectory points and the desired sampling interval, we determine the number of missing trajectory points and use a placeholder token [m] to indicate them, and formulate the preprocessed sparse trajectory $\mathcal{T}^\epsilon$.

To obtain the road condition of the missing trajectory point $s$, we identify its nearest observed forward and backward trajectory points, $s_f$ and $s_b$, respectively. For instance, suppose we have a missing point $s$ timestamped at 8:00:45, its observed forward point $s_f = p_1$, and the backward point $s_b = p_2$. Then, we use Equation 2 to calculate the road condition of point $s$

## H  The Detailed Description of Variants

- **PLMTrajRec - Randomly initialized BERT:** We randomly initialize the parameters of BERT instead of using the pre-trained BERT based on large-scale corpus datasets.
- **PLMTrajRec - GPT-2:** We replace the pretrained BERT with GPT-2 [24].
- **PLMTrajRec - Llama:** We replace the pretrained BERT with Llama [44].
- **w/o IF-guided explicit trajectory prompt:** We remove the IF-guided explicit trajectory prompt.
- **w/o AF-guided implicit trajectory prompt:** We use the '[MASK]' token in the BERT to represent the missing location.
- **w/o dual trajectory prompts:** We remove both the IF-guided explicit and implicit trajectory prompt, and the missing points are represented by the '[MASK]' token in BERT.
- **w/o reference tokens:** We remove the trajectory feature transformation layer in the trajectory embedder module.

## I  Scalability Results on Different Datasets

### I.1  Scalability Results on Chengdu-Large

As shown in Table 7, our model consistently achieves superior performance on the Chengdu-Large dataset under varying settings. Notably, under three different sampling intervals, the model surpasses

Table 7: Scalability analysis. The performance comparison on the Chengdu-Large dataset when trained with different data ratios. Red denotes the best result, and blue denotes the second-best result.

| Setting | Data Ratio | 20% | | 40% | | 60% | | 80% | | 100% | |
|---|---|---|---|---|---|---|---|---|---|---|---|
| | Metric | Acc(%) | RMSE | Acc(%) | RMSE | Acc(%) | RMSE | Acc(%) | RMSE | Acc(%) | RMSE |
| $\mu = 4$ minutes | MTrajRec | 65.58 | 874.5 | 66.73 | 852.2 | 68.29 | 842.5 | 69.32 | 837.4 | 70.09 | 828.5 |
| | T3s + Decoder | 66.63 | 863.4 | 68.10 | 838.1 | 68.84 | 825.0 | 69.79 | 815.4 | 70.62 | 771.3 |
| | T2vec + Decoder | 66.39 | 883.2 | 69.02 | 849.0 | 69.30 | 831.9 | 70.82 | 813.9 | 71.49 | 792.8 |
| | RNTrajRec | 68.38 | 825.1 | 71.11 | 806.6 | 71.89 | 788.9 | 72.77 | 772.6 | 73.62 | 764.8 |
| | MM-STGED | 71.28 | 781.3 | 73.76 | 753.7 | 74.89 | 739.6 | 75.33 | 727.6 | 75.51 | 718.9 |
| | **PLMTrajRec** | **77.57** | **482.6** | **78.04** | **461.9** | **78.53** | **446.2** | **78.82** | **429.7** | **78.97** | **421.6** |
| $\mu = 2$ minutes | MTrajRec | 74.92 | 638.1 | 77.13 | 593.8 | 78.48 | 572.7 | 79.27 | 554.7 | 80.09 | 528.6 |
| | T3s + Decoder | 74.41 | 614.9 | 77.09 | 574.9 | 77.82 | 566.4 | 78.78 | 526.9 | 79.19 | 499.5 |
| | T2ev + Decoder | 76.39 | 593.7 | 78.19 | 533.2 | 78.86 | 528.4 | 79.53 | 502.7 | 80.11 | 461.4 |
| | RNTrajRec | 75.19 | 584.4 | 77.47 | 502.0 | 79.26 | 473.4 | 79.98 | 433.0 | 80.95 | 389.8 |
| | MM-STGED | 79.34 | 474.2 | 81.20 | 449.7 | 82.37 | 384.0 | 83.10 | 366.9 | 83.77 | 337.7 |
| | **PLMTrajRec** | **83.92** | **336.9** | **84.47** | **312.7** | **84.82** | **292.5** | **84.98** | **285.9** | **85.11** | **273.0** |
| $\mu = 1$ minute | MTrajRec | 78.51 | 438.0 | 81.18 | 386.2 | 82.36 | 370.0 | 82.82 | 359.7 | 83.75 | 347.7 |
| | T3s + Decoder | 78.38 | 447.2 | 80.17 | 364.1 | 82.10 | 335.6 | 82.83 | 312.9 | 83.11 | 299.8 |
| | T2ev + Decoder | 77.18 | 432.7 | 79.66 | 408.4 | 81.29 | 384.5 | 82.14 | 335.0 | 82.35 | 301.8 |
| | RNTrajRec | 80.51 | 374.0 | 82.37 | 331.9 | 83.72 | 300.1 | 84.03 | 275.8 | 84.21 | 253.0 |
| | MM-STGED | 83.02 | 283.1 | 84.17 | 265.8 | 85.32 | 244.2 | 85.98 | 219.7 | 86.21 | 203.9 |
| | **PLMTrajRec** | **88.13** | **174.6** | **88.84** | **159.9** | **89.63** | **150.3** | **89.82** | **144.7** | **89.94** | **137.5** |

the SOTA baseline trained on the full dataset while using only 20% of the data. It achieves an improvement of 3.76% in Acc and 33.78% in RMSE. These results indicate that our model is well-suited for the recovery task in data-scarce scenarios, demonstrating its strong generalization capability.

## I.2 Scalability Results on Porto-Small

Table 8: Scalability analysis. The performance comparison on the Porto-Small dataset when trained with different data ratios. Red denotes the best result, and blue denotes the second-best result.

| Setting | Data Ratio | 20% | | 40% | | 60% | | 80% | | 100% | |
|---|---|---|---|---|---|---|---|---|---|---|---|
| | Metric | Acc(%) | RMSE | Acc(%) | RMSE | Acc(%) | RMSE | Acc(%) | RMSE | Acc(%) | RMSE |
| $\mu = 4$ minutes | MTrajRec | 49.55 | 733.3 | 51.10 | 652.1 | 51.77 | 638.1 | 52.09 | 629.2 | 52.36 | 590.1 |
| | T3s + Decoder | 50.34 | 715.8 | 51.09 | 648.3 | 51.67 | 632.7 | 52.01 | 628.7 | 52.24 | 594.9 |
| | T2vec + Decoder | 50.46 | 700.4 | 51.81 | 656.1 | 52.36 | 641.4 | 52.72 | 620.6 | 53.13 | 571.0 |
| | RNTrajRec | 51.41 | 677.8 | 52.66 | 636.2 | 53.38 | 583.0 | 53.94 | 562.1 | 54.59 | 549.1 |
| | MM-STGED | 55.62 | 642.1 | 56.09 | 601.4 | 56.68 | 574.6 | 57.04 | 539.2 | 57.30 | 510.4 |
| | **PLMTrajRec** | **56.54** | **432.9** | **56.90** | **410.5** | **57.07** | **396.5** | **57.14** | **384.8** | **57.61** | **376.9** |
| $\mu = 2$ minutes | MTrajRec | 57.62 | 629.3 | 59.39 | 562.1 | 60.25 | 529.0 | 61.10 | 486.2 | 61.65 | 451.5 |
| | T3s + Decoder | 57.84 | 617.3 | 59.62 | 573.8 | 60.14 | 541.7 | 61.29 | 483.2 | 61.75 | 461.2 |
| | T2vec + Decoder | 57.57 | 603.4 | 60.00 | 555.6 | 60.61 | 532.8 | 61.94 | 475.5 | 62.24 | 438.0 |
| | RNTrajRec | 60.60 | 587.8 | 62.25 | 518.5 | 62.91 | 464.5 | 63.11 | 448.7 | 63.39 | 433.9 |
| | MM-STGED | 62.24 | 479.7 | 63.72 | 452.5 | 64.07 | 437.3 | 64.78 | 414.3 | 65.69 | 400.8 |
| | **PLMTrajRec** | **65.01** | **371.4** | **65.47** | **348.4** | **65.76** | **327.0** | **66.17** | **310.7** | **66.40** | **294.6** |
| $\mu = 1$ minute | MTrajRec | 67.26 | 499.0 | 69.39 | 414.6 | 70.69 | 386.4 | 71.25 | 352.3 | 71.65 | 332.3 |
| | T3s + Decoder | 68.10 | 482.1 | 69.28 | 421.4 | 70.47 | 372.5 | 71.04 | 335.9 | 71.78 | 328.0 |
| | T2vec + Decoder | 67.89 | 514.3 | 69.69 | 428.7 | 70.82 | 392.9 | 71.10 | 349.0 | 71.86 | 334.7 |
| | RNTrajRec | 69.66 | 419.8 | 71.17 | 368.2 | 71.93 | 352.2 | 72.19 | 336.3 | 72.31 | 325.7 |
| | MM-STGED | 71.54 | 382.6 | 72.05 | 364.5 | 72.64 | 355.1 | 72.84 | 338.8 | 73.16 | 321.9 |
| | **PLMTrajRec** | **72.28** | **316.6** | **73.06** | **264.9** | **73.57** | **244.4** | **74.08** | **221.9** | **74.42** | **211.7** |

The experimental results on the Porto-Small dataset are presented in Table 8. Our model significantly outperforms all baseline models. We attribute this to the model's effective use of world knowledge stored in the PLM, which enhances the model's ability to understand the characteristics of trajectory data. When compared with state-of-the-art trajectory recovery models such as MM-STGED and RNTrajRec, our model achieves improvements of 14.43% and 17.56%, respectively.

## I.3 Scalability Results on Porto-Large

The experimental results on the Porto-Large dataset are presented in Table 9, our model consistently achieves the best performance across all evaluation metrics. Compared with the state-of-the-art model MM-STGED, PLMTrajRec outperforms it by margins of 14.84%, 11.57%, 11.65%, 10.46%, and 10.85% under data ratios of 20%, 40%, 60%, 80%, and 100%, respectively. Remarkably, even when trained on only 20% of the data, our model exceeds the performance of the baseline trained on the full dataset. These results highlight the strong scalability of PLMTrajRec.

Table 9: Scalability analysis. The performance comparison on the Porto-Large dataset when trained with different data ratios. Red denotes the best result, and blue denotes the second-best result.

| Setting | Data Ratio | 20% | | 40% | | 60% | | 80% | | 100% | |
|---|---|---|---|---|---|---|---|---|---|---|---|
| | Metric | Acc(%) | RMSE | Acc(%) | RMSE | Acc(%) | RMSE | Acc(%) | RMSE | Acc(%) | RMSE |
| $\mu = 4$ minutes | MTrajRec | 58.41 | 473.1 | 59.47 | 439.3 | 60.74 | 392.2 | 61.13 | 378.2 | 61.87 | 362.1 |
| | T3s + Decoder | 59.54 | 459.3 | 61.92 | 411.4 | 62.50 | 398.1 | 62.92 | 381.8 | 63.52 | 375.1 |
| | T2vec + Decoder | 59.30 | 449.7 | 62.19 | 408.8 | 63.45 | 382.8 | 63.98 | 361.9 | 64.24 | 358.4 |
| | RNTrajRec | 61.66 | 419.4 | 64.07 | 352.6 | 64.69 | 339.7 | 65.24 | 328.8 | 65.98 | 311.0 |
| | MM-STGED | 63.82 | 364.7 | 65.11 | 328.3 | 65.79 | 313.7 | 66.37 | 302.1 | 66.66 | 296.3 |
| | **PLMTrajRec** | **66.53** | **311.6** | **66.89** | **295.6** | **67.12** | **279.8** | **67.38** | **271.0** | **67.57** | **264.2** |
| $\mu = 2$ minutes | MTrajRec | 62.12 | 418.9 | 64.58 | 372.0 | 65.10 | 359.9 | 66.04 | 332.4 | 66.85 | 320.5 |
| | T3s + Decoder | 63.49 | 391.7 | 64.68 | 366.7 | 66.22 | 348.6 | 66.79 | 335.9 | 67.39 | 324.9 |
| | T2vec + Decoder | 63.94 | 402.4 | 66.70 | 352.2 | 67.21 | 320.6 | 68.41 | 316.9 | 68.84 | 308.3 |
| | RNTrajRec | 64.19 | 388.0 | 67.33 | 340.0 | 67.81 | 331.2 | 68.30 | 304.8 | 68.88 | 286.6 |
| | MM-STGED | 67.89 | 315.3 | 68.29 | 288.6 | 68.80 | 276.2 | 69.71 | 264.5 | 70.03 | 255.9 |
| | **PLMTrajRec** | **71.82** | **242.6** | **72.37** | **221.4** | **72.88** | **210.4** | **73.02** | **202.7** | **73.14** | **191.9** |
| $\mu = 1$ minute | MTrajRec | 69.10 | 359.1 | 71.48 | 318.9 | 71.98 | 264.0 | 72.31 | 238.2 | 72.63 | 216.5 |
| | T3s + Decoder | 68.82 | 366.3 | 70.90 | 327.6 | 70.89 | 271.6 | 71.11 | 244.4 | 71.31 | 225.9 |
| | T2vec + Decoder | 69.43 | 337.1 | 71.29 | 300.1 | 72.01 | 252.5 | 72.38 | 221.3 | 72.54 | 208.1 |
| | RNTrajRec | 70.19 | 305.0 | 71.48 | 258.1 | 71.92 | 215.3 | 72.51 | 199.8 | 72.72 | 185.8 |
| | MM-STGED | 71.97 | 264.6 | 72.50 | 187.5 | 72.95 | 174.7 | 73.58 | 159.0 | 74.03 | 149.6 |
| | **PLMTrajRec** | **76.29** | **157.6** | **76.72** | **138.6** | **76.91** | **129.5** | **77.17** | **125.3** | **77.30** | **116.7** |

# J   Case Study

We conduct a case study on the Chengdu-Small dataset to visualize the trajectory recovery performance with various baselines. As shown in Figure 7, we draw the truth and recovered trajectory, where red points represent the truth trajectory points and blue points indicate the recovered trajectory points. We find that the recovered trajectory aligns well with the road network, demonstrating the effectiveness of using road segment and moving rate to represent trajectory point. To facilitate a more intuitive comparison of recovery performance among different models, we focus on two distinct regions, labeled as A and B, for visual analysis. In region A, characterized by a relatively simple road network structure, all models can accurately recover road segments. Among them, PLMTrajRec maps trajectory points better by leveraging PLM. In contrast, region B exhibits a more intricate road network with multiple accessible routes. RNTrajRec captures spatial-temporal correlations of trajectories that are not adequate and recovers incorrect road segments. PLMTrajRec not only excels in road segment recovery but also in accurately matching the actual trajectory points. This success can be attributed to its ability to model missing trajectory points through the implicit trajectory prompt, thereby introducing more valuable information and improving performance.

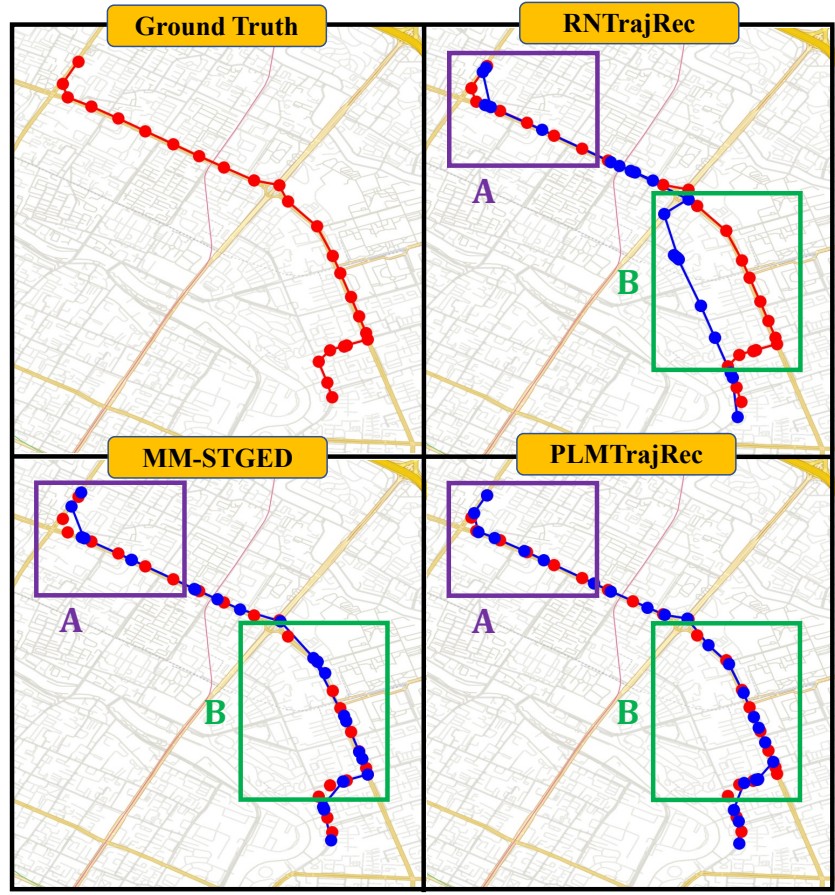

Figure 7: Case study on the Chengdu-Small dataset. Red points represent the truth trajectory points and blue points represent the recovered trajectory points.

