# OpenReview forum: "PLMTrajRec: A Scalable and Generalizable Trajectory Recovery Method with Pre-trained Language Models"
_NeurIPS.cc/2025/Conference — NeurIPS 2025 spotlight_

### Official Review · Reviewer_VuDY · 2025-06-25

**Clarity:** 3
**Significance:** 3
**Originality:** 2
**Rating:** 5
**Confidence:** 4

**Summary:**

The paper introduces a novel trajectory recovery model PLMTrajRec to recover map-matched trajectory by fine-tuning a  pre-trained Language Model. By converting sampling intervals, movement features, and traffic flow into token sequences, the model unifies sparse trajectories of heterogeneous sampling rates and predicts both road-segment IDs and within-segment moving ratios. Experiments on four public datasets show outstanding gains of accuracy and RMSE compared to SOTA methods.

**Questions:**

See weakness.

**Ethical Concerns:**

["NO or VERY MINOR ethics concerns only"]

**Final Justification:**

I carefully reviewed the authors’ rebuttal and the additional discussion. The new large-scale Xi’an dataset strengthens the performance.  Although zero-shot transfer across distinct road networks is technically infeasible with the current output space, the authors acknowledge this limitation and outline a plausible path toward topology-independent embeddings. Given the original strengths, I maintain an accept rating (5) and believe the paper meets the NeurIPS bar.

**Limitations:**

Yes, the authors discuss the limitations in Appendix A.

**Paper Formatting Concerns:**

None.

**Quality:**

3

**Strengths And Weaknesses:**

Strengths

(1) The work focuses on an interesting problem, extending pretrained language models to trajectory recovery and showing that generic PLMs's ability in spatio-temporal tasks.

(2) This paper delivers a comprehensive evaluation on four real-world datasets and shows promising performance.

(3) The architecture including the interval-aware embedder, dual prompts, and LoRA fine-tuning is clearly illustrated, and every module is justified through ablation studies.



Weaknesses

(1) The demonstration of third challenge (Figure 1) is confusing.

(2) Experimentation on more cities could be considered.

(3) Apart from the temporal granularity, the generalization ability of the method can be considered for zero-shot experiments between different cities.

---

> ### Author Rebuttal · Authors · 2025-07-30
>
> Thank you for your insightful feedback! We are grateful for your acknowledgment of our task settings and novelty. Below, we address each of your concerns and questions in detail:
>
> > [W1] The demonstration of third challenge (Figure 1) is confusing.
>
> We apologize for the confusion caused by the third challenge illustrated in Figure 1. We would like to offer a clearer explanation:
>
> The third challenge emphasizes the importance of capturing traffic conditions for both observed and missing trajectory points to ensure accurate recovery. For example, a vehicle may stop-and-go in congested zones, while moving faster in uncongested ones.
> From Figure 1, we can determine that the user may take route $R\_1$ based on the road conditions of observed points $p\_1$ and $p\_4$ (Figure 1 (a); however, we cannot determine how the user travels within route $R\_1$, such as whether they accelerate, decelerate, or drive at a constant speed. This information needs to be further extracted based on the road conditions of the missing points, $p\_2$ and $p\_3$. However, since the locations of these missing points are unknown, we cannot obtain their actual road condition information. Therefore, it is difficult to accurately infer the trajectory's movement. To address it, we propose a Road Condition Passing Mechanism, as shown in Lines 208-217. We will revise the figure caption and corresponding text to clarify.
>
>
>
> > [W2] Experimentation on more cities could be considered.
>
> We have added a Xi'an trajectory dataset covering an area of 8.20 km × 7.98 km, which contains 3,392 road segments and 400,000 trajectories. The scale of the dataset is larger than Chengdu-Large. It is divided into training, validation, and test sets using a 7:1:2 ratio. The model settings remain consistent with those used for the other datasets, as described in the experimental settings. We conduct trajectory recovery experiments using a sparse sampling interval of 2 minutes, aiming to recover trajectories at 15-second intervals.
>
> |                  | Acc       | Recall    | Prec      | MAE       | RMSE      |
> | ---------------- | --------- | --------- | --------- | --------- | --------- |
> | RNTrajRec        | 70.58     | 70.63     | 71.38     | 166.3     | 274.3     |
> | MM-STGED         | 71.16     | 71.80     | 72.19     | 153.9     | 254.7     |
> | PLMTrajRec (our) | **73.29** | **74.11** | **75.28** | **117.4** | **173.1** |
>
> The results show that our model outperforms the state-of-the-art map-constrained trajectory recovery methods, MM-STGED and RNTrajRec, on the Xi'an dataset in terms of both road segment recovery accuracy and spatial distance error. Specifically, our model improves accuracy by 2.13% and reduces RMSE by 81.6 meters. These results highlight the effectiveness of our model in trajectory recovery.
>
>
>
> > [W3] Apart from the temporal granularity, the generalization ability of the method can be considered for zero-shot experiments between different cities.
>
>
>
> We appreciate this thoughtful concern. Zero-shot transfer between cities would indeed be valuable. However, as stated in Appendix A.1, PLMTrajRec's key limitation is its reliance on fixed road segment IDs for output classification. This design couples the model to a specific road topology, making direct application to different road networks technically infeasible.
>
> While PLMTrajRec cannot achieve zero-shot transfer across road networks, it supports few-shot transfer through its scalability and generalization capabilities. Our experiments (Table 2 and Appendix I, Tables 7-9) show the model maintains competitive performance with only 20% of training data, demonstrating robustness in few-shot scenarios.
>
> In future work, we plan to develop universal or topology-independent road segment embeddings that decouple the output space from specific road networks, enabling broader generalization across different road networks or evolving regions.

---

> > ### Comment · Reviewer_VuDY · 2025-08-04
> >
> > Thanks for your reply! I have no further questions, as I've given a very positive score, I decide to maintain "Accept".

---

> ### Author Response · Authors · 2025-08-06
>
> Thank you for taking the time to review our rebuttals and for your positive score. We are pleased that our responses have effectively addressed your questions.

---

### Official Review · Reviewer_huba · 2025-06-30

**Clarity:** 3
**Significance:** 3
**Originality:** 2
**Rating:** 5
**Confidence:** 3

**Summary:**

This paper introduces a method entitled PLMTrajRec which tries to recover missing points in sparse trajectories.

There are three challenges in this task: 1) lack of datasets; 2) different sampling intervals of input trajectory data are hard to recover due to complex and dynamic spatiotemporal traffic conditions; 3) extracting road information for missing points is challenging.

To tackle the three problems, the authors 1) use pretrained language models (PLM) as backbone which equipped with human common sense and relieve the lack of data; 2) introduce an interval and feature-guided (IF-guided) explicit traejctory prompt to provide the sampling interval information to PLM, and also use a "Trajectory Interval Unification" module to gaurantee that each two points in the input trajectory has a same sampling interval; 3) design an area flow-guided (AF-guided) implicit trajectory prompt to provide dynamic traffic flow information for PLM, and also a road condition passing mechanism to retrieve relevant road condition for missing points to be recovered.

The authors test their model on four datasets, and the results show PLMTrajRec and its variant gets the best score on the metrics of Accuracy, Recall, Precision, Mean Absolute Error (MAE), and Root Mean Square Error (RMSE).

The authors conducted a solid ablation study to verify the effectiveness of each module. Besides, the authors also provide experiments to contrast the zero-shot ability between PLMTrajRec and other methods. Finally, the authors also provide efficient analysis to show computation  costs of the model.

**Questions:**

1.Questions
a.Line 197, you mentioned a function $f(d_{s, l})$ which seems to map the distance between a point and a road segment to a number (larger distance, smaller value of the number). So, does the function actually function as the weight to obtain $h_s^{road}$? And why use a \textbf{ransomly initialized vector} (line 200)? Is the vector fixed or leaned?
b.Line 223-226. You mentioned you use "self-attention", but $H'$ acts as the query, and $E_w$ serves as both key and vlue. In my understanding of attention, you may use cross attention instead of self-attention.
c.Line 228 and Figure 2. You said you incorporated the Transformer position embedding PE into \textbf{each} element of $Z$. 1) I hope you can provide more information about the position encoding you used. Is PE something about "Learnable Fourier Features" you mentioned in Appendix F? 2) In Figure 2, it seems that PE is a token instead of auxiliary embeddings for each token.
d. Line 208-211. You only collect information for two points (the nearest observed forward and backward trajectory points), what do you think of extracting the information of the road between the two points? Will that be helpful?
2.Writing Suggestions
a.The format should be consistent in section Preliminaries. The first words of Line 99, 103, 107, 113.
b.More visualization and information about your method is helpful for understanding. Although the authors provide visualization in appendix, I do not see any link to Appendix K. I hope the author can add some links to the paper so that readers can see the visualization of the model result. I also suggest the author can provide an example of raw prompt and LLM output in appendix, that helps readers understand your tasks.

**Ethical Concerns:**

["NO or VERY MINOR ethics concerns only"]

**Final Justification:**

I believe the quality of this paper is good, and the rebuttal has addressed the issues I raised. I will raise my score to accept.

**Limitations:**

yes

**Quality:**

3

**Strengths And Weaknesses:**

1.Quality: This paper proposed a new method, provided detailed information and background of method. The author also provided code for reproducing. Also, the author provides rich and solid experiments to verify the rationality and effectiveness of the proposed method.
2.Clarity: The paper is well organized, and provide link to appendix in each place that need extra information. However, in some place, the logic maybe not clarified.
3.Significance: This paper tries to propose a method to solve the problem of recovering missing points in trajectories. The model can be applied in ride-hailing applications, provide more accuracy information for consumers and administrator. The results on the test set show the potential to utilize this method.
4.Originality: Using PLM for Spatiotemporal data-related downstrained tasks is not a new thing. The authors should further elaborate on the common methods in related fields and clarify the innovations of this study.

---

> ### Author Rebuttal · Authors · 2025-07-30
>
> Thank you for your insightful feedback! We are grateful for your acknowledgment of our task settings and insightful analysis. Below, we address each of your concerns and questions in detail:
>
> > [W] The authors should further elaborate on the common methods in related fields and clarify the innovations of this study.
>
> While PLMs have been explored in certain spatiotemporal applications, their use in trajectory recovery remains limited. Some existing LLM-based models for trajectory-related tasks, such as TrajCogn [1] and Mobility-LLM [2], primarily focus on extracting movement intentions from existing trajectory features to better understand and represent the trajectory. However, in the trajectory recovery task, only a subset of trajectory points is observed, making it challenging to infer rich movement patterns or spatiotemporal semantics from such sparse information. Moreover, these models typically adopt a decoder-only architecture for trajectory learning. As demonstrated in our ablation experiments, using a decoder-only structure (e.g., GPT-2 or LLaMA) results in reduced recovery accuracy. This is because trajectory recovery requires access to both preceding and succeeding context simultaneously. In contrast, decoder-only models follow a left-to-right autoregressive design, which limits them to leveraging only past information, failing to incorporate future context.
>
> Existing trajectory recovery approaches primarily focus on capturing trajectory spatiotemporal features for improving accuracy. For instance, MTrajRec leverages GRU-based sequential encoders and autoregressive decoding to recover trajectories. RNTrajRec emphasizes modeling the spatial interactions between trajectory points and the road network using a Transformer architecture. MM-STGED incorporates semantic features by encoding trajectories as spatiotemporal graphs. Although these methods perform well, they generally assume access to abundant training data with consistent sampling intervals.
>
> However, it is challenging to obtain large-scale, high-quality trajectories in real-world situations. Therefore, our key innovation is explicitly designed to address two critical but underexplored challenges: (1) recovering trajectories with limited training data, and (2) generalizing to trajectories with varying sampling intervals. And we address them by leveraging PLMs’ broad general knowledge to mitigate the lack of domain-specific supervision. This enables our model to effectively handle data-scarce and sparse scenarios. Our experimental results support this claim, as shown in Tables 2, 3, 7, 8, and 9.
>
>
>
> [1] Zhou Z, Lin Y, Wen H, et al. TrajCogn: Leveraging LLMs for Cognizing Movement Patterns and Travel Purposes from Trajectories[J]. arXiv preprint arXiv:2405.12459, 2024.
>
> [2] Gong L, Lin Y, Lu Y, et al. Mobility-llm: Learning visiting intentions and travel preference from human mobility data with large language models[J]. Advances in Neural Information Processing Systems, 2024, 37: 36185-36217.
>
>
>
> > [Q1.a] Line 197, you mentioned a function $f(d_{s,l})$,which seems to map the distance between a point and a road segment to a number (larger distance, smaller value of the number). So does the function actually function as the weight to obtain $h_s^{road}$. And why use a randomly initialized vector (line 200)? Is the vector fixed or leaned?
>
> Yes, the function $f(d_{s,l})$ represents the weights of trajectory points to road segment $l$ according to their distance, and these weights are used to calculate $\mathbf{h}\_s^{\text{road}}$. For a trajectory point $s$, the calculation formula of $\mathbf{h}\_s^{\text{road}}$ is:
>
> $
> \mathbf{h}\_s^{\mathrm{road}} =  \frac{ {\sum\_{l=1}^{|\mathcal{V} |}} f(d\_{s,l}) \cdot \mathbf{M}_l  }{\sum\_{l=1}^{|\mathcal{V} |} f(d\_{s,l})}
> $
>
> where $\mathbf{M}_l \in \mathbb{R}^F$ is the learnable embedding of road segment $l$ , and $|\mathcal{V}|$ is the total number of road segments.
>
> The randomly initialized vector in line 200 is $\mathbf{M}_l$. It is randomly initialized at the beginning and then learned during model training. We will clarify this in the revised version.
>
>
>
> > [Q1.b] Line 223-226. You mentioned you use "self-attention", but $\textbf{H}'$' acts as the query, and $\textbf{E}$ serves as both key and vlue. In my understanding of attention, you may use cross attention instead of self-attention.
>
> You are correct. It is cross attention instead of self-attention. We will modify this in the revised version.
>
>
>
> > [Q1.c] Line 228 and Figure 2. You said you incorporated the Transformer position embedding PE into \textbf{each} element of $Z$. 1) I hope you can provide more information about the position encoding you used. Is PE something about "Learnable Fourier Features" you mentioned in Appendix F? 2) In Figure 2, it seems that PE is a token instead of auxiliary embeddings for each token.
>
> We apologize for the confusion caused by Figure 2. The positional encoding (PE) mentioned in Line 228 and Figure 2 refers to the standard positional encoding mechanism used in Transformer architectures. Our implementation adds PE directly to each token embedding in $Z$, consistent with conventional Transformer usage.
>
> Formally, the positional encoding is defined as:
> $$
> \mathrm{PE}\_{(pos,\,2i)} = \sin \left (\frac{pos}{10000^{2i/d\_{\mathrm{model}}}}\right), \quad
> \mathrm{PE}\_{(pos,\,2i+1)} = \cos \left (\frac{pos}{10000^{2i/d\_{\mathrm{model}}}}\right),
> $$
> where $pos$ denotes the position index in the sequence, $i$ is the dimension index, and $d_{\text{model}}$ is the model's embedding dimension (512 in our paper). $\text{PE}(pos,k)$ represents the value at the $k$-th dimension for the token at position $pos$.
>
> This is distinct from the learnable Fourier features described in Appendix F, which encode raw spatial coordinates (latitude and longitude) of trajectory points, as detailed in Section 4.2.2 Trajectory Feature Extractor. We will correct this in the revised version.
>
>
>
> > [Q1.d] Line 208-211. You only collect information for two points (the nearest observed forward and backward trajectory points), what do you think of extracting the information of the road between the two points? Will that be helpful?
>
>
> We appreciate the reviewer’s thoughtful concern. To evaluate the effect of incorporating road network information between the two observed trajectory points, we conducted an additional experiment on Chengdu-Small with sampling interval of 2 minutes. In this variant, we constructed a representation of the missing point by weighting the road network features surrounding the two observed points.
>
> |                                  | Acc   | Recall | Prec  | MAE   | RMSE  |
> | -------------------------------- | ----- | ------ | ----- | ----- | ----- |
> | with road network                | 81.68 | 84.23  | 88.15 | 191.2 | 373.5 |
> | PLMTrajRec (with road condition) | 81.76 | 84.31  | 88.38 | 187.4 | 366.2 |
>
> As shown in Table above, this modification led to slightly degraded performance compared to our original approach. We believe this is due to feature redundancy, as the road network information of the surrounding trajectory points has already been encoded during the trajectory point embedding stage (as shown in Line 195-202). Reintroducing similar road features may lead to overlapping or redundant signals, which in turn may hinder effective learning. Moreover, trajectories are inherently dynamic, and movement patterns in the same location can vary significantly across different times of day. In contrast, the road network is static and does not capture temporal variations. Therefore, we argue that incorporating time-dependent traffic road condition information is more effective than relying solely on static road network between points.
>
>
> > [Q2.a] The format should be consistent in section Preliminaries. The first words of Line 99, 103, 107, 113
>
> Thank you for your careful observation. We will modify them in the revised version.
>
>
> > [Q2.b] Add more visualization of the model result and provide an example of raw prompt and LLM output in appendix.
>
> Thanks for your suggestions. We will add more visualization examples.
> For the example of raw prompt and LLM output, we have the following additions:
>
> Consider a sparse trajectory $\mathcal{T} = \langle p_1, \cdots, p_{15} \rangle  $ with 15 trajectory points with a sampling interval of 4 minutes, starting at 8 o’clock on Monday and ending at 9 o’clock on Monday. And the total movement distance is 40 kilometers. Our goal is to recover it within a sampling interval of 15 seconds; therefore, there are 240 output trajectory points (4 points per minute, totaling 60 minutes). According to this trajectory, our prompts are:
> Task Part: Sparse trajectory recovery. Target Part: Output the road segment and moving ratio for each point in the trajectory. Content Part: The sparse trajectory is sampled on average four minutes and aims to recover trajectory every fifteen seconds. Time Part: The trajectory started at eight o’clock on Monday and ended at nine o’clock on  Monday. Movement Part: Total time cost: sixty minutes zero seconds. Total space transfer distance: 40 kilometers.
>
> For the output of the model, it should output the corresponding road segment ID $e_i$ and moving rate $r_i$ for each trajectory point, where $1\le i\le 240$​.

---

### Official Review · Reviewer_pB4W · 2025-07-03

**Clarity:** 3
**Significance:** 3
**Originality:** 3
**Rating:** 4
**Confidence:** 3

**Summary:**

This paper addresses the problem of recovering missing points in sparse spatiotemporal trajectories, which is crucial for traffic-related applications but challenged by sparse data, varying sampling intervals, and difficulty in modeling dynamic road conditions. The authors propose PLMTrajRec, a trajectory recovery method that leverages a pre-trained language model (PLM) to overcome these challenges. Their contributions include: (1) Designing dual trajectory prompts (explicit and implicit) that encode sampling intervals, movement features, and road conditions into formats understandable by a PLM. (2) Introducing an interval-aware trajectory embedder that standardizes trajectories with diverse sampling intervals. (3) Proposing a road condition passing mechanism to estimate road conditions at missing points. Extensive experiments on four real-world datasets demonstrate the method’s effectiveness, scalability, and generalization, outperforming existing baselines, especially in low-data and zero-shot settings.

**Questions:**

1. The ablation compares different PLMs but does not deeply explore why encoder-only PLMs outperform others for this task. Could the authors elaborate on how task-specific needs (e.g., bidirectionality, context length) drive this choice, and how sensitive the performance is to PLM size and architecture?

2. The paper demonstrates good scalability to low-data settings, but at the cost of increased inference time and memory. Are there scenarios (e.g., edge devices, real-time applications) where this trade-off becomes prohibitive? Could the authors discuss potential ways to further reduce the footprint?

3. The design of the dual prompts is clever, but it’s not clear how sensitive the method is to the exact prompt format and content. Have the authors explored alternative prompt formulations, or can they recommend guidelines for constructing effective prompts in similar tasks?

**Ethical Concerns:**

["NO or VERY MINOR ethics concerns only"]

**Final Justification:**

The current rebuttal is sufficient to resolve my main reservations. Since I had already given a positive score, I am inclined to maintain my current rating.

**Limitations:**

See in weakness

**Quality:**

3

**Strengths And Weaknesses:**

Strengths
1. This work tackles an important and practical problem with real-world implications in transportation systems, urban planning, and mobility analytics.

2. The paper is technically solid, proposing several thoughtful innovations (e.g., dual prompts, interval-aware embedding, and road condition passing) that are well-motivated and clearly explained.

3. The paper is clearly written and well-organized, making the methodology and experiments easy to follow.

Weaknesses
1. The method introduces additional computational and memory overhead compared to some baselines, which is mitigated but not fully eliminated by LoRA fine-tuning.

2. Some details, such as the choice of PLM (why BERT over GPT or Llama) and sensitivity to the choice of PLM, are discussed but could benefit from deeper empirical analysis.

3. While the paper discusses broader impacts in the appendix, the main text could highlight more explicitly potential societal risks (e.g., privacy, misuse of mobility data).

---

> ### Author Rebuttal · Authors · 2025-07-30
>
> Thank you for your insightful feedback! We are grateful for your acknowledgment of our task settings and novelty. Below, we address each of your concerns and questions in detail:
>
> > [W1 & Q2] The method introduces additional computational and memory overhead compared to some baselines, and discussing potential ways to further reduce inference time and memory.
>
>
>
> While PLMTrajRec has a relatively large parameter count, LoRA fine-tuning mitigates this issue. Only approximately 35% of parameters are trainable while the rest remain frozen, making the effective model size increase modest compared to full fine-tuning. The performance gains justify this trade-off in most settings. To further reduce overhead, we deliberately select a lightweight PLM architecture. As described in Appendix D.2, we use BERT-small with only 4 Transformer layers and a hidden size of 512. This significantly reduces inference time and memory footprint, making it suitable for resource-constrained settings.
>
> We acknowledge that PLM-based models still incur overhead that could be prohibitive in extreme low-resource scenarios. Several optimization techniques can address this: (1) Model quantization, converting floating-point parameters (FP32) to lower-bit integers (INT8/INT4) or removing less important weights/neurons to reduce parameter count and computation; (2) Knowledge distillation, where a lightweight student model is co-trained to mimic the PLM teacher's trajectory representations, enabling efficient inference while maintaining generalization. These strategies show great potential for extending PLMTrajRec to wider deployment scenarios.
>
>
>
> > [W2 & Q1] More discussion about the choice of PLM and sensitivity to the choice of PLM.
>
>
>
> We thank you for this helpful suggestion. Our choice of encoder-based PLM architecture is motivated by the trajectory recovery task's requirements. Recovering missing trajectory points requires understanding both past and future context. BERT's bidirectional encoding is well-suited for this need, allowing the model to jointly consider information from both directions. In contrast, decoder-only PLMs like GPT and LLaMA process tokens unidirectionally, relying only on left-to-right context. While advantageous for autoregressive generation, this limits their ability to incorporate future information, which is critical for accurate trajectory recovery. Our ablation studies (as shown in Table 4) show PLMTrajRec consistently outperforms decoder-only PLMs under identical training conditions.
>
> Regarding model size sensitivity, we experimented with larger BERT variants (BERT-base and BERT-large). Performance gains were limited relative to the increase in parameter count and inference cost. BERT-small, with only 4 layers and 512-dimensional hidden states, offers a favorable performance-efficiency trade-off, making it suitable for training and deployment in low-resource settings.
>
>
>
> > [W3] While the paper discusses broader impacts in the appendix, the main text could highlight more explicitly potential societal risks (e.g., privacy, misuse of mobility data).
>
> The trajectory recovery task enables data collectors to store and transmit only sparsely sampled or partially masked trajectories, mitigating privacy exposure. Using PLMTrajRec to recover dense trajectories when needed reduces the amount of sensitive raw data retained or exposed. Nevertheless, any system capable of reconstructing dense mobility data carries misuse risks, such as surveillance or unauthorized profiling. We recommend deployment only in controlled environments with full anonymization.
>
>
>
> > [Q3] The design of the dual prompts is clever, but it’s not clear how sensitive the method is to the exact prompt format and content. Have the authors explored alternative prompt formulations, or can they recommend guidelines for constructing effective prompts in similar tasks?
>
> In our pre-experiments, we explored several prompt formulation variants on Chengdu dataset with sampling interval of 2 minutes. Initially, prompts included only the task objective and expected output format (Task Part and Target Part). We then gradually added contextual information: sampling intervals of sparse and dense trajectories (Context Part), and trajectory-specific characteristics like time span and total movement distance (Time Part and Movement Part). This incremental enhancement led to our final prompt formulation. As shown below, the experiment results demonstrate that semantically richer content consistently improved performance, indicating that informative prompts more effectively activate PLM reasoning abilities.
>
> |                                                       | Acc   | Recall | Prec  | MAE   | RMSE  |
> | ----------------------------------------------------- | ----- | ------ | ----- | ----- | ----- |
> | base (prompt with Task and Target Part)               | 81.47 | 84.07  | 87.94 | 197.3 | 379.2 |
> | base + Context Part                                   | 81.66 | 84.20  | 88.17 | 191.7 | 371.0 |
> | PLMTrajRec (base + Context Part + trajectory Feature) | 81.76 | 84.31  | 88.38 | 187.4 | 366.2 |
>
> Based on these observations, effective prompt design should align with task structure and semantic context. Specifically, effective prompts should: 1) Clearly define the task goal (Task Part); 2) Describe input conditions like sampling sparsity (Context Part); 3) Provide relevant trajectory statistics such as time range and movement distance (Time Part and Movement Part); and 4) Specify desired output representation like full trajectories or road segment IDs (Target Part). Integrating these elements helps PLMs better understand both input and expected output, improving performance. This structured design may generalize to other related tasks.

---

### Official Review · Reviewer_mGi6 · 2025-07-03

**Clarity:** 3
**Significance:** 2
**Originality:** 3
**Rating:** 4
**Confidence:** 2

**Summary:**

This paper proposes PLMTrajRec, a novel method for recovering missing points in sparse trajectory data using Pre-trained Language Models (PLMs). It leverages the scalability of PLMs and structured prompts to generalize across various sampling intervals and limited dense data. The model incorporates Dual Trajectory Prompts and an Interval-aware Trajectory Embedder, which unifies trajectories to a fixed interval and extracts features using learnable spatial encodings and road condition estimation. A message-passing-inspired mechanism is used to infer road conditions at missing points using nearby observed points. Experiments on four public datasets demonstrate superior performance, scalability, and generalization compared to baselines.

**Questions:**

Please refer to the weakness section.

**Ethical Concerns:**

["NO or VERY MINOR ethics concerns only"]

**Final Justification:**

I believe the rebuttal has properly addressed my concerns, and since there is consensus among the reviewers, I will maintain my original rating.

**Limitations:**

yes

**Quality:**

3

**Strengths And Weaknesses:**

Strengths

- **Novel Integration of PLM for Map-Matched Trajectory Recovery**: The attempt to apply Pre-trained Language Models (PLMs) to the map-matched trajectory recovery task is highly novel. Unlike prior works that mainly rely on sequence models or graph-based approaches, this paper leverages the powerful sequence modeling capabilities and pretrained knowledge of language models, which is an original idea.
- **Thoughtful Design for Key Challenges**: The methodology is well-designed to address three core challenges: data scarcity, varying sampling intervals, and road condition inference.
    - PLM scalability mitigates the limited data problem.
    - An interval-aware embedder and explicit prompts effectively handle various sampling intervals (generalizability).
    - Implicit prompts and the road condition passing mechanism enable inference of road conditions at missing points.

    These components are logically integrated and well-motivated.

- **Reproducibility**: The authors provide code and dataset access, enhancing the reproducibility of the results.
- **Comprehensive Experimental Evaluation**:
    - **Multiple Datasets & Scales**: The model is evaluated on four public datasets with different city types and scales, showing its general performance.
    - **Various Sampling Intervals**: Experiments cover three different sampling intervals (4min, 2min, 1min), systematically validating the model’s ability to handle varying temporal sparsity.
    - **Targeted Studies**: The paper includes well-designed experiments to highlight core strengths, such as scalability studies (training with limited data) and zero-shot studies (testing on unseen intervals).

Weaknesses

- **Lack of Deep Justification for PLM’s “Language” Capability in Trajectory Tasks**: The paper may lack a deeper explanation of *why* PLMs are effective for trajectory recovery beyond their strength as general sequence encoders. It would be helpful to analyze what aspects of trajectory data align with PLMs’ language modeling abilities—do the prompts facilitate any kind of linguistic reasoning, or are they merely embedding manipulators? Currently, the model may be perceived as simply utilizing PLM as a strong sequence encoder without fully exploiting its “language understanding” capabilities.
- **Dependence on Road Network Structure**: Since the model’s output is directly tied to road segment IDs and segment-level positioning, it raises concerns about the model’s dependence on a specific road network structure. For example:
    - How would the model perform on regions with entirely different road network structures (as hinted at in Appendix A.1)?
    - What happens if the road network evolves over time?

    There is potential for significant performance degradation in such cases.

    Additionally, it’s worth questioning whether the core ideas of PLMTrajRec—such as PLM usage, trajectory prompts, and interval-aware embedding—can be transferred to **free-space trajectory recovery** settings, where road network information is not available. Some parts of the model (e.g., output layer design and certain feature extraction steps) seem tightly coupled to road network structure. It would be valuable to hear the authors’ perspective on the adaptability of PLMTrajRec to free-space scenarios.

---

> ### Author Rebuttal · Authors · 2025-07-30
>
> Thank you for your insightful feedback! We are grateful for your acknowledgment of our task settings and novelty. Below, we address each of your concerns and questions in detail:
>
>
>
> > [W1.1] The paper may lack a deeper explanation of why PLMs are effective for trajectory recovery beyond their strength as general sequence encoders.
>
> While trajectories are not conventional language texts, they share substantial structural similarities with natural language that make PLMs well-suited for trajectory recovery. A trajectory can be analogized to a sentence, where each trajectory point corresponds to a word. The spatiotemporal dependencies between trajectory points mirror the contextual dependencies between words in a sentence. Movement patterns such as turning and acceleration correspond to semantic relationships between words. Trajectory recovery resembles a cloze-style task in NLP, where missing tokens must be inferred from surrounding context. These parallels enable PLMs to leverage their pre-trained knowledge and reasoning capabilities to extract trajectory semantics, enhancing generalization in sparse data scenarios.
>
>
>
> > [W 1.2] It would be helpful to analyze what aspects of trajectory data align with PLMs’ language modeling abilities—do the prompts facilitate any kind of linguistic reasoning, or are they merely embedding manipulators? Currently, the model may be perceived as simply utilizing PLM as a strong sequence encoder without fully exploiting its “language understanding” capabilities.
>
> Given the similarities between trajectories and natural language, we use natural language prompts to activate the PLM's reasoning abilities for trajectories. Our IF-guided explicit trajectory prompts inform the PLM about the task objective, expected output format, and key trajectory features including start/end times and total movement distance. These prompts provide task-oriented descriptions that guide trajectory recovery, functioning beyond simple token manipulation. The AF-guided implicit prompts encode road condition dynamics that influence vehicle movement—for example, vehicles decelerate in congested areas and move steadily in free-flowing regions. PLMs can leverage their pre-trained temporal and behavioral priors to interpret this information.
>
> Moreover, since PLMs are not inherently trained for trajectory data, we introduce learnable reference tokens $\textbf{E}_w$ in Section 4.2.3 to bridge the gap between natural language and structured trajectory features. These tokens facilitate semantic alignment and enable PLMs to better integrate spatiotemporal representations into their language-centric framework.
>
> Our ablation studies (as shown in Table 4) confirm the importance of both trajectory prompts and reference tokens; removing them significantly degrades performance.
>
>
>
> > [W2] The model’s dependence on a specific road network structure. And what happens if the model transfer to entirely different road network structures, or the road network evolves over time.
>
> We appreciate this important concern. In map-matched trajectory recovery, modeling trajectory and road network interactions is crucial for accurate recovery. Like prior work including MTrajRec [22], RNTrajRec [5], and MM-STGED [33], our model utilizes road network structure to better understand how trajectories move through the map.
>
> As we stated in Appendix A.1, PLMTrajRec's key limitation is its reliance on fixed road segment IDs for output classification. This design couples the model to a specific road topology, making it technically infeasible to directly apply a trained model to the different road networks or evolved network structures. While PLMTrajRec cannot achieve zero-shot transfer across road networks, it supports few-shot transfer through its scalability and generalization capabilities. Our experiments (Table 2 and Appendix I, Tables 7-9) show the model maintains competitive performance with only 20% of training data, demonstrating robustness in few-shot scenarios. In practice, this enables quick adaptation to changed or evolved road networks.
>
> In future work, we plan to develop universal or topology-independent road segment embeddings that decouple the output space from specific road networks, enabling broader generalization across different road networks or evolving regions.
>
>
>
> > [Q] The adaptability of PLMTrajRec to free-space scenarios.
>
> We appreciate this thoughtful question. The core ideas of PLMTrajRec can indeed be transferred to free-space trajectory recovery. Free-space trajectory recovery faces similar challenges: limited dense trajectory data and the need to generalize across varied sampling intervals. PLMTrajRec specifically addresses these challenges through PLMs and interval-aware embeddings, which naturally extend to free-space settings. Our prompt design encodes task instructions, sampling intervals, and trajectory features (temporal duration and movement distance) that are independent of road network constraints and transferable to free-space recovery.
>
> PLMTrajRec requires only minimal architectural modifications for free-space trajectory recovery. The key difference lies in output format: map-matched recovery outputs road segment IDs and moving rates, while free-space recovery outputs GPS coordinates. This is accommodated by modifying only the output layer, replacing road segment IDs and moving rates with latitude/longitude coordinates, and changing the loss function to coordinate-based MSE loss. Our experiments (as shown in Table 1) show that combining free-space models like TERI and TrajBERT with our map-constrained output layer yields good results.
>
> While both approaches improve trajectory data quality, we believe map-constrained recovery is more practical for real-world applications. Vehicles typically travel along roads, and map-constrained recovery directly reflects vehicle behavior on the road network. Free-space recovery requires additional map-matching steps to relate predicted points back to the road network.

---

### Comment · Area_Chair_CqMx · 2025-08-07
**Rebuttal**

Hi Reviewers,

Can you please discuss whether the rebuttal changes your opinion or ratings? Thanks!

---

### Note · Authors · 2025-08-12

We sincerely thank the AC and all reviewers for their time and valuable feedback. **We are grateful that our novelty, motivation, problem setting, and solid experimental results have been acknowledged, with all reviewers giving positive scores.** We appreciate the opportunity to present and clarify our work again.

## Summary of Contributions
To the best of our knowledge, this is the first work to simultaneously address scalability and sampling interval generalization in trajectory recovery. Leveraging the general knowledge of PLMs, our method can accurately recover trajectories from limited data while generalizing to various sampling intervals, effectively mitigating the scarcity of dense trajectories and the need for repeated training.

Technically, we bridge the gap between sparse trajectory and PLMs, enabling effective interpretation of spatiotemporal information. We propose dual-trajectory prompts, road condition passing mechanism, and interval–aware embedder to identify sampling intervals, capture trajectory features, and model spatiotemporal correlations.

## Broader Impact
As the first application of PLMs to trajectory recovery, our work offers new insights for related fields, reduces dependence on large-scale datasets, mitigates privacy risks, and lowers training costs.

## Key Points in Rebuttal
1. We demonstrate PLM effectiveness for trajectory data by analogy between trajectory recovery and cloze tasks (Reviewer mGi6). We show that existing PLM-based works for spatiotemporal tasks are limited in sparse trajectory recovery due to severe information loss in sparse trajectory (Reviewer huba), and we address this with a dedicated design. We find that encoder-based PLMs with bidirectional encoding outperform decoder-based models for recovery tasks (Reviewer pB4W).

2. Our method, using only 20% of the data, adapts well to other cities (Reviewer mGi6, VuDY) and can handle free-space trajectory recovery with minimal output-layer changes (Reviewer mGi6).

3. We summarize guidelines for trajectory prompt design and strategies for deploying PLMs for extremely low-resource scenarios (Reviewer pB4W).

## Future Revision Commitment
We will further refine the writing and clarify details in the final version.

## Closing Acknowledgement
Once again, we sincerely thank the AC and all reviewers for their thoughtful feedback and support, which has greatly improved the clarity and rigor of our work.

---

### Decision · Program_Chairs · 2025-09-17

**Decision:**

Accept (spotlight)

**Comment:**

The paper received two borderline accept and two accept final recommendations. Reviewers agreed that applying a pre-trained language model for trajectory recover in traffic related applications is novel and valid. The authors provided a rebuttal. One reviewer raised the score to accept after reading the rebuttal. AC recommends to accept the paper